# *Lin28a/let-7* pathway modulates the *Hox* code via *Polycomb* regulation during axial patterning in vertebrates

Tempei Sato[1,2,3], Kensuke Kataoka[1,3], Yoshiaki Ito[1,4], Shigetoshi Yokoyama[2,5], Masafumi Inui[2,6], Masaki Mori[1,7], Satoru Takahashi[8], Keiichi Akita[9], Shuji Takada[2], Hiroe Ueno-Kudoh[2,10], Hiroshi Asahara[1,2,11,12]*

[1]Department of Systems BioMedicine, Graduate School of Medical and Dental Sciences, Tokyo Medical and Dental University, Tokyo, Japan; [2]Department of Systems BioMedicine, National Research Institute for Child Health and Development, Tokyo, Japan; [3]Research Fellow of Japan Society for the Promotion of Science, Tokyo, Japan; [4]Research Core, Tokyo Medical and Dental University, Tokyo, Japan; [5]Laboratory of Metabolism, National Institutes of Health, Bethesda, United States; [6]Laboratory of Animal Regeneration Systemology, Meiji University, Kanagawa, Japan; [7]Department of Medical Chemistry, Shiga University of Medical Science, Shiga, Japan; [8]Department of Anatomy and Embryology, University of Tsukuba, Ibaraki, Japan; [9]Department of Clinical Anatomy, Graduate School of Medical and Dental Sciences, Tokyo Medical and Dental University, Tokyo, Japan; [10]Reproduction Center, Yokohama City University, Yokohama, Japan; [11]AMED-CREST, Japan Agency for Medical Research and Development (AMED), Tokyo, Japan; [12]Department of Molecular Medicine, The Scripps Research Institute, La Jolla, United States

*For correspondence:
asahara@scripps.edu

Competing interests: The authors declare that no competing interests exist.

**Abstract** The body plan along the anteroposterior axis and regional identities are specified by the spatiotemporal expression of *Hox* genes. Multistep controls are required for their unique expression patterns; however, the molecular mechanisms behind the tight control of *Hox* genes are not fully understood. In this study, we demonstrated that the *Lin28a/let-7* pathway is critical for axial elongation. *Lin28a*[−/−] mice exhibited axial shortening with mild skeletal transformations of vertebrae, which were consistent with results in mice with tail bud-specific mutants of Lin28a. The accumulation of *let-7* in *Lin28a*[−/−] mice resulted in the reduction of PRC1 occupancy at the *Hox* cluster loci by targeting *Cbx2*. Consistently, Lin28a loss in embryonic stem-like cells led to aberrant induction of posterior *Hox* genes, which was rescued by the knockdown of *let-7*. These results suggest that the *Lin28/let-7* pathway is involved in the modulation of the '*Hox* code' via *Polycomb* regulation during axial patterning.

## Introduction

The precise positioning of each organ and tissue has to be tightly controlled during embryogenesis. The body plan along the anteroposterior axis is modulated by the spatiotemporal expression of *Hox* genes, which is known as the '*Hox* code' (**Wellik, 2007**; **Mallo and Alonso, 2013**). Hox genes encode a family of transcription factors that contain a helix-turn-helix type homeodomain. In vertebrates, Hox genes are organized into four paralogous clusters (A to D) that can be divided into thirteen groups. The members of each paralogous group have partially redundant functions, but also acquire independent functions (**Wellik, 2007**; **Mallo et al., 2010**). During development, the

paralogous group at the 3' end of the clusters is expressed in the anterior part of the body, while the more 5' genes are expressed in the more posterior part, towards the tail (*Deschamps and van Nes, 2005*; *Dressler and Gruss, 1989*; *Duboule and Dollé, 1989*; *Gaunt and Strachan, 1996*; *Graham et al., 1989*; *Izpisúa-Belmonte et al., 1991b*; *Izpisúa-Belmonte et al., 1991a*). These expression patterns modulate the anterior and posterior axis and specify the regional anatomical identities of the vertebrae: Hox gene-knockout mice show anterior transformations where specific vertebrae mimic the morphology of a more anterior one (*Chisaka and Capecchi, 1991*; *Chisaka et al., 1992*; *Le Mouellic et al., 1992*; *Condie and Capecchi, 1993*; *Jeannotte et al., 1993*; *Small and Potter, 1993*; *Davis and Capecchi, 1994*; *Horan et al., 1994*; *Horan et al., 1995b*; *Horan et al., 1995a*; *Kostic and Capecchi, 1994*; *Davis et al., 1995*; *Rancourt et al., 1995*; *Suemori et al., 1995*; *Boulet and Capecchi, 1996*; *Fromental-Ramain et al., 1996b*; *Fromental-Ramain et al., 1996a*; *Carpenter et al., 1997*; *Chen and Capecchi, 1997*; *Chen et al., 1998*; *Manley and Capecchi, 1997*; *van den Akker et al., 2001*; *Garcia-Gasca and Spyropoulos, 2000*; *Wahba et al., 2001*; *Wellik and Capecchi, 2003*; *McIntyre et al., 2007*). Multistage controls, such as transcriptional, posttranscriptional, and epigenetic regulation, are required for the nested expression patterns of *Hox* genes (*Mallo and Alonso, 2013*).

As for epigenetic control, *Polycomb* group (PcG) genes are involved in *Hox* gene regulation via the chromatin architecture at *Hox* cluster loci in a developmental time-dependent manner (*Soshnikova, 2014*). PcG genes form two complexes, the *Polycomb* Repressive Complex (PRC) one and PRC2. PRC2 includes Ezh2, which can catalyze H3K27me3 at target loci, and consequently, this specific histone modification causes the recruitment of PRC1 via Cbx2 in the complex to silence gene expression. Thus, the accumulation of PcG complexes at *Hox* clusters during embryogenesis leads to the transcriptional silencing of *Hox* genes, which is supported by evidence that the ablation of PcG genes dysregulates *Hox* gene expression, resulting in subsequent skeletal transformation in anteroposterior patterning (*Mallo and Alonso, 2013*; *Soshnikova, 2014*). During embryogenesis, PcG gene expression gradually diminishes (*Hashimoto et al., 1998*), which leads to the initiation of spatiotemporal *Hox* gene expression. However, the molecular mechanisms underlying the termination of PcG gene expression remain largely unclear.

Previously, we generated a whole-mount in situ hybridization database called 'EMBRYS' that covers ~1600 transcription factors and RNA-binding factors using mice at embryonic day (E)9.5, E10.5, and E11.5 (*Yokoyama et al., 2009*). Among these data, we were particularly interested in dynamic expressional changes of *Lin28a* during embryogenesis: at E9.5, *Lin28a* is expressed ubiquitously, whereas its expression gradually diminishes from head to tail at E10.5 and E11.5 (*Yokoyama et al., 2008*; *Yokoyama et al., 2009*). These unique expressional changes prompted us to analyze if *Lin28a* is involved in the spatiotemporal regulation of *Hox* genes.

*Lin-28* was identified as a heterochronic gene that regulates the developmental timing of multiple organs in *Caenorhabditis elegans* (*C.elegans*) (*Moss et al., 1997*). *Lin-28* encodes an RNA-binding protein, and the loss of function of *Lin-28* causes precocious development, with skipping of events that are specific to the second larval stage (*Ambros and Horvitz, 1984*; *Moss et al., 1997*). In contrast, mutants of *let-7*, a microRNA-encoding heterochronic gene, exhibit reiteration of the fourth larval developmental stage because of failures in terminal differentiation and cell-cycle exit (*Pasquinelli et al., 2000*; *Reinhart et al., 2000*). Importantly, *Lin-28* and *let-7* form a negative feedback loop that is essential for developmental timing in *C. elegans*. This reciprocal regulation between Lin28a and *let-7* is well conserved in mammals (*Moss and Tang, 2003*; *Viswanathan et al., 2008*); Lin28a promotes the degradation of *let-7* precursors (*Heo et al., 2009*; *Chang et al., 2013*), whereas *let-7* inhibits *Lin28a* expression via posttranscriptional regulation (*Moss and Tang, 2003*).

Vertebrates possess two homologs of *Lin28* genes, *Lin28a* and *Lin28b*. *Lin28a* is highly expressed in pluripotent stem cells and is ubiquitously expressed in the early embryonic stage, and its expression is diminished during development (*Yang and Moss, 2003*; *Shyh-Chang and Daley, 2013*; *Yokoyama et al., 2008*; *Yokoyama et al., 2009*). In contrast, *Lin28b* is dominantly expressed in testes, placenta, and fetal liver, as well as in undifferentiated hepatocarcinoma (*Guo et al., 2006*). The versatile functions of *Lin28a* are observed in diverse events, such as germ layer formation (*Faas et al., 2013*), germ cell development (*West et al., 2009*), neural development (*Yang et al., 2015*), glucose metabolism (*Zhu et al., 2011*), and skeletal development (*Aires et al., 2019*; *Robinton et al., 2019*; *Papaioannou et al., 2013*). Conversely, *let-7*-family genes are highly expressed in differentiated tissues, and their products function as tumor suppressors via the

inhibition of oncogenes such as *c-Myc*, *K-ras*, and *Hmga2* (*Mayr et al., 2007*; *Lee and Dutta, 2007*; *Johnson et al., 2005*; *Sampson et al., 2007*). These observations prompted us to test the potential 'heterochronic' function of *Lin28a* in vertebrate development; however, it remains largely unclear if the evolutionarily fundamental function of the *Lin-28* and *let-7* negative feedback loop in the regulation of developmental timing and pattern of *C. elegans* is conserved or adapted in vertebrates.

In this work, we generated *Lin28a* knockout (*Lin28a⁻/⁻*) mice and analyzed the function of this gene in developmental patterning. We showed that the *Lin28a*/*let-7* pathway is critical for axial elongation and vertebral patterning. *Lin28a⁻/⁻* mice exhibited axial shortening with mild skeletal transformations of vertebrae, which were consistent with results observed in mice with tail bud-specific gain/loss of function of Lin28a (*Aires et al., 2019*; *Robinton et al., 2019*). The accumulation of *let-7*-family microRNAs in *Lin28a⁻/⁻* mice resulted in the reduction of PRC1 occupancy at the *Hox* cluster loci by targeting *Cbx2*. Consistent with these results, Lin28a loss in embryonic stem-like cells led to the aberrant induction of posterior *Hox* genes, which was rescued by knockdown of *let-7*-family microRNAs. These results suggest the involvement of the *Lin28*/*let-7* pathway in the modulation of the '*Hox* code' in vertebrates.

## Results

### *Lin28a⁻/⁻* mice exhibit skeletal patterning defects

*Lin28a* exhibits unique spatiotemporal expression changes during early development (*Figure 1A*; *Yang and Moss, 2003*; *Yokoyama et al., 2008*; *Yokoyama et al., 2009*). At E9.5, *Lin28a* is expressed ubiquitously; and subsequently, its expression disappears from head to tail at around E10.5 and E11.5 (*Yokoyama et al., 2008*; *Yokoyama et al., 2009*). To examine the potential significance of these dynamic expression changes and of the developmental function of *Lin28a* in mice, we generated *Lin28a⁻/⁻* mice (*Figure 1—figure supplement 1*). The normal Mendelian ratio of genotypes was observed for *Lin28a⁻/⁻* mice during early to mid embryogenesis. However, the frequency of *Lin28a⁻/⁻* mice decreased from E17.5 and after birth. Most of the *Lin28a⁻/⁻* mice died perinatally or within a few days after birth (*Supplementary file 1*). *Lin28a⁻/⁻* mice exhibited short stature compared with wild-type (Wt) mice and showed severe growth defects (*Figure 1—figure supplement 2*). These findings are consistent with previous reports that *Lin28a* is necessary for normal growth (*Shinoda et al., 2013*); however, our *Lin28a⁻/⁻* mice showed severe phenotypes that might have been caused by differences in gene targeting construct and genetic background.

We then examined anteroposterior axis formation in *Lin28a⁻/⁻* mice since *Lin28a⁻/⁻* mice showed a slight anterior shift of the hindlimbs and shortened tails (*Figure 1B*). To define the details of these phenotypes, whole-mount in situ hybridization of *Myog* and *Fgf8* was performed to outline somites and limb buds. The hindlimbs of *Lin28a⁻/⁻* mice shifted anteriorly by one somite (from the 23rd to the 28th expression domains of *Myog*), whereas the position of the forelimb buds of *Lin28a⁻/⁻* mice were not altered (*Figure 1C*). These results are supported by previous reports that tail bud-specific overexpression or knockout of Lin28a affects the number of caudal vertebrae (*Aires et al., 2019*; *Robinton et al., 2019*).

To analyze the potential functions of *Lin28a* in skeletal patterning, Alcian blue and Alizarin red S staining were applied to the skeletal preparations. Although bone and cartilage development was normal, skeletal patterning defects were observed in *Lin28a⁻/⁻* mice (*Figure 1D–H*). In *Lin28a⁻/⁻* mice, the anterior arch of the atlas was formed from the second cervical vertebra (C2) (*Figure 1D*), and not from C1, as normally observed, or from the fusion of C1 and C2 (*Figure 1E*). These transformations were observed in 64.3% of *Lin28a⁻/⁻* mice and in 21.1% of *Lin28a⁺/⁻* mice; in contrast, they were never found in Wt mice (*Table 1*). There were only six pairs of true ribs attached to the sternum in *Lin28a⁻/⁻* mice, whereas Wt and *Lin28a⁺/⁻* mice had seven pairs (*Figure 1F*). Furthermore, an abnormal number of ribs was observed in *Lin28a⁻/⁻* mice at 100% penetrance, whereas Wt and *Lin28a⁺/⁻* mice exhibited the normal 13 pairs of ribs (*Figure 1G* and *Table 1*). These results suggest that posterior transformations of vertebral identity occur in the 7th and 13th thoracic vertebrae during skeletal patterning. Moreover, partial transformations were observed in the first sacral vertebra (S1), producing a morphological feature of lumbar vertebrae on only one side (*Figure 1H*). The frequency of these observations was significantly higher in *Lin28a⁻/⁻* mice (*Table 1*). Finally, *Lin28a⁻/⁻*

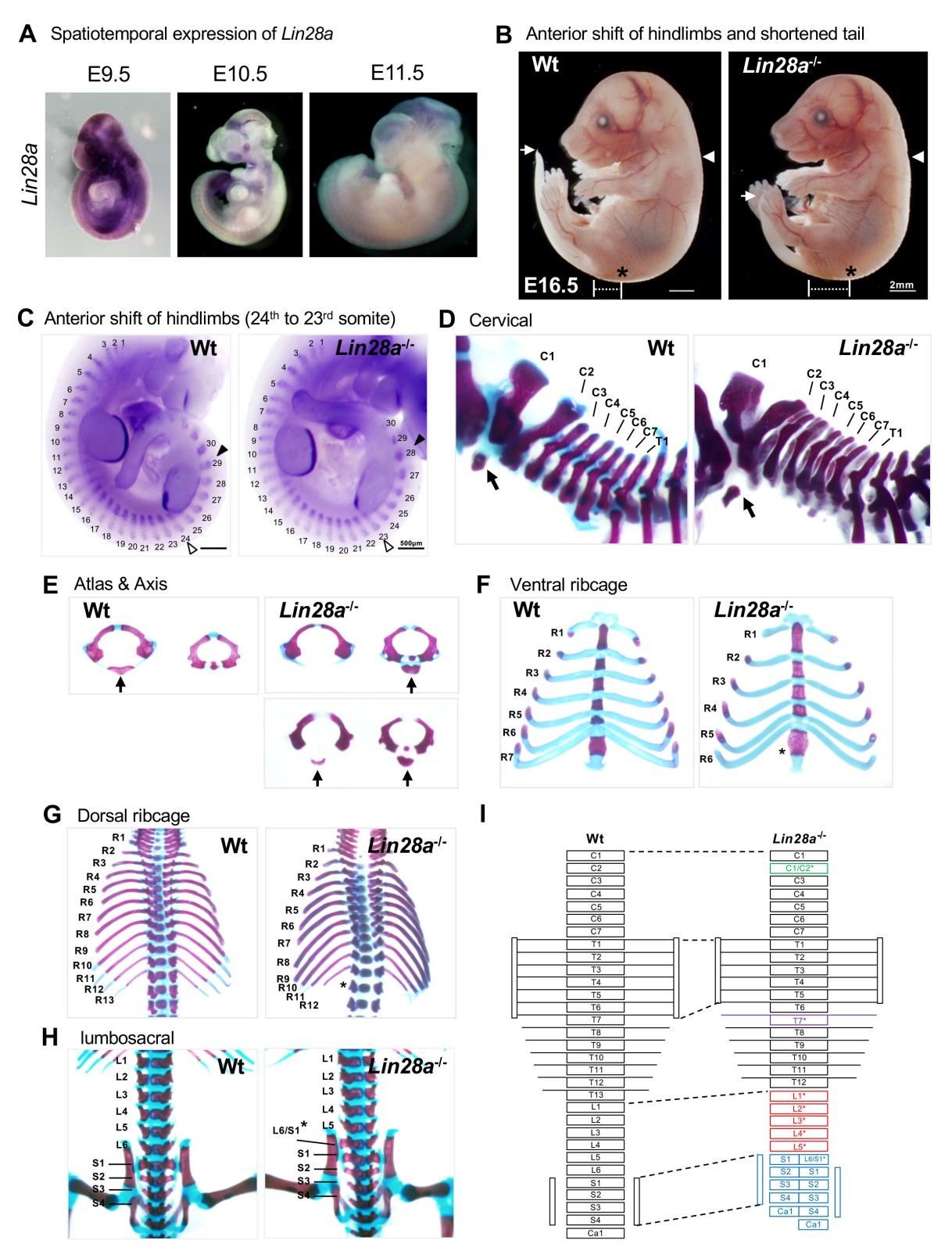

**Figure 1.** Skeletal patterning defects in *Lin28a$^{-/-}$* mice. (**A**) Whole-mount in situ hybridization of *Lin28a* in E9.5–11.5 embryos. (**B**) Lateral views of Wt (left panel) and *Lin28a$^{-/-}$* mice (right panel) at E16.5. White arrow, the tip of the tail; white arrowhead, forelimb position; asterisk, hindlimb position. (**C**) Whole-mount in situ hybridization of *Myog* and *FGF8* in E10.5 embryos. The numbers indicate the expression domains of *Myog*. White arrowhead, the starting position of the hindlimb bud; black arrowhead, the ending position of the hindlimb bud. (**D–H**) Representative skeletal preparations of Wt (left

*Figure 1 continued on next page*

*Figure 1 continued*

panels) and *Lin28a*$^{-/-}$ mice (right panels). Abbreviations/marks are described below. Lateral views of cervical and upper thoracic vertebrae (D); anterior views of the atlas and the axis (E); ventral views of the ribcage (F); dorsal views of thoracic vertebrae and ribs (G); and dorsal views of lumbar and sacral vertebrae (H) are shown. (I) Schematic diagram of skeletal phenotypes in *Lin28a*$^{-/-}$ mice. Each abbreviation in (D–I) indicates as follows: C1–C7, 1$^{st}$ to 7$^{th}$ cervical vertebrae; T1-T13, 1$^{st}$ and 13$^{th}$ thoracic vertebrae; R1–R13, 1$^{st}$ to 13$^{th}$ ribs; L1–L6, 1$^{st}$ to 6$^{th}$ lumbar vertebrae; S1–S4, 1$^{st}$ to 4$^{th}$ sacral vertebrae; Ca1, 1$^{st}$ caudal vertebrae. Black arrows in (D–E) indicate anterior arch of the atlas. Asterisks in (F–I) indicate the sites where skeletal deformations occurred.

The online version of this article includes the following figure supplement(s) for figure 1:

**Figure supplement 1.** Generation of *Lin28a*$^{-/-}$ mice.
**Figure supplement 2.** *Lin28a*$^{-/-}$ mice exhibit growth defects.

mice showed various skeletal transformations (*Figure 1I*), suggesting that *Lin28a* plays a critical role in the specification of vertebrae along the anteroposterior axis.

### *Hox* genes are dysregulated in *Lin28a*$^{-/-}$ mice

The morphologies and characteristics of each vertebra are specified by the spatiotemporal expression of *Hox* genes (*Wellik, 2007*). It was remarkable that *Lin28a*$^{-/-}$ mice exhibited global transformations with high penetration, whereas mutants of *Hox* genes showed abnormalities in a limited region of vertebrae. To test if *Hox* genes are involved in the phenotypes found in *Lin28a*$^{-/-}$ mice, we examined *Hox* gene expression during embryogenesis. Quantitative real-time polymerase chain reaction (q-PCR) analyses of *Hox* genes were performed at E9.5, a time at which *Lin28a* was ubiquitously expressed in Wt mice (*Figure 1A*). *Lin28a*$^{-/-}$ mice exhibited global dysregulation of *Hox* genes, which was most remarkable for the 5' (posterior) *Hox* genes (*Figure 2A*). Whole-mount in situ hybridization analyses revealed that the expression domain of *Hoxc13* and *Hoxd12* was enlarged anteriorly (*Figure 2B and C*, and *Figure 2—figure supplement 1*). In contrast, there were no significant changes in the expression domain of the other *Hox* genes upregulated in *Lin28a*$^{-/-}$ mice (Hoxa3, d3, b8, c8, a11, and a13) (*Figure 2—figure supplement 1*). These results suggest that the short-tailed phenotype in *Lin28a*$^{-/-}$ mice might be caused by the anteriorization of *Hox* paralogous group 12 and 13 expression.

We next focused on the skeletal patterning defects from the cervical to sacral region. It is known that Hox genes are modulated by retinoic acid (RA) signaling, and that RA exposure causes posterior transformations of vertebrae via global anteriorization of *Hox* gene expression (*Kessel and Gruss, 1991*). Since Hox genes were dysregulated in *Lin28a*$^{-/-}$ embryos, we hypothesized that the patterning defects of vertebrae observed in *Lin28a*$^{-/-}$ mice are caused by the perturbation of *Hox* gene expression. To test this, we investigated the effects of perturbation of Hox gene expression by RA on skeletal pattern formation in Lin28a mutants. RA was injected intraperitoneally at 7.5 days postcoitum (dpc) and the skeletal patterning of each fetus was analyzed. We found that *Lin28a* mutant embryos showed RA sensitivity. *Lin28a*$^{+/-}$ mice that received RA treatment showed loss of the 13$^{th}$

**Table 1.** Summary of skeletal abnormalities in *Lin28a* mutant mice.

| | Anterior arch of the atlas[*] | Ribs[†] | | Sternum attachment[‡] | | | Lumbar[§] | | |
| --- | --- | --- | --- | --- | --- | --- | --- | --- | --- |
| | | 13 | 12 | 7 | 61 | 6 | 5 | | L6/S1[*] |
| Wt (n = 16) | 0 | 16 (100%) | 0 | 16 (100%) | 0 | 7 (43.8%) | 8 (50%) | | 1 (6.2%) |
| Lin28a$^{+/-}$ (n = 19) | 4 (21.1%) | 19 (100%) | 0 | 19 (100%) | 0 | 0 | 18 (94.7%) | | 1 (5.3%) |
| Lin28a$^{-/-}$ (n = 14) | 9 (64.3%) | 0 | 14 (100%) | 0 | 14 (100%) | 0 | 9 (64.3%) | | 5 (35.7%) |

The percentages of each phenotype are shown in parenthesis.

* The anterior arch of the atlas was formed from C2 or via fusion.

† Total number of pairs of ribs.

‡ Total number of pairs of true ribs that were attached to the sternum.

§ Total number of lumbar vertebrae. L6/S1* indicates an abnormal sacral vertebra that had morphological features of a lumbar vertebra on only one side.

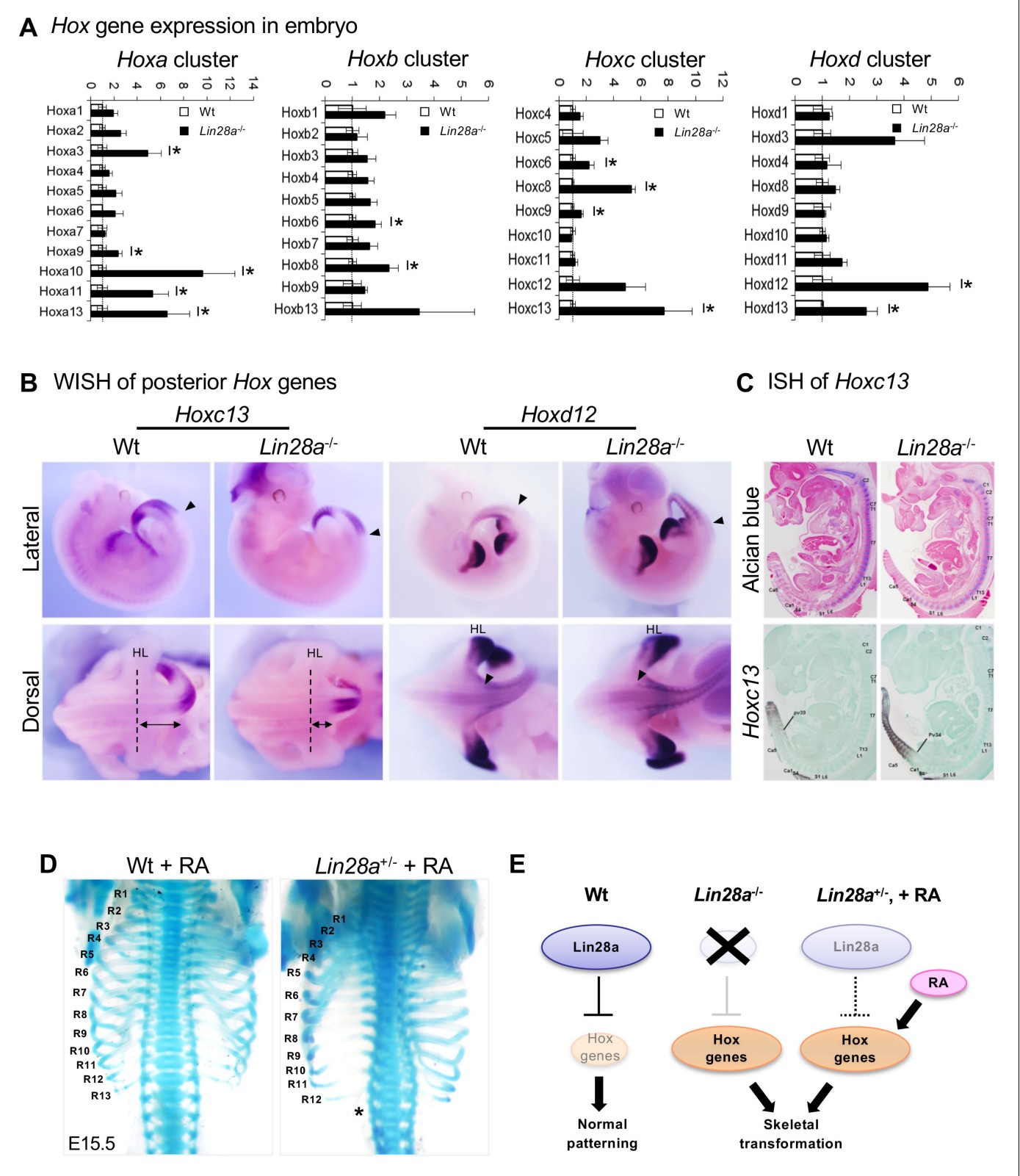

**Figure 2.** *Hox* gene dysregulation in *Lin28a*<sup>-/-</sup> mice. (**A**) q-PCR analyses of all *Hox* genes. All data are expressed as the mean ± standard error of the mean (SEM) (n = 3). *p<0.05. (**B**) Whole-mount in situ hybridization of *Hox* genes in E11.5 embryos. Lateral views (top panels) and dorsal views (bottom panels) of hindlimb and tail region are shown. Black arrowhead, anterior domain of *Hox* gene; HL, hindlimb; dashed line, hindlimb position; two-way arrow, distance from the hindlimb to the anterior domain of *Hoxc13*. (**C**) Histological analysis of E12.5 animals. Alcian blue staining (top panels) and in situ

*Figure 2 continued on next page*

*Figure 2 continued*

situ hybridization of *Hoxc13* (bottom panels) are shown. (D) Skeletal preparations of Wt (left panel) and *Lin28a*[+/−] mice (right panel) that received RA treatment. R1–R13, 1[st] to 13[th] ribs; asterisk, the ablation of the 13[th] rib. See also *Figure 2—figure supplement 2*. (E) Summary of *Hox* gene dysregulation in *Lin28a* mutants.

The online version of this article includes the following source data and figure supplement(s) for figure 2:

**Source data 1.** Source data related to panel (A).
**Figure supplement 1.** Whole-mount in situ hybridization of *Hox* genes in Lin28 knockout embryos (Related to Fig.
**Figure supplement 2.** RA sensitivity in *Lin28a* mutant mice.

pair of ribs, which coincided with the findings observed in *Lin28a*[−/−] mice, whereas no obvious defects were observed in Wt littermates (*Figure 2D*). In contrast, no additional defects in the thoracic region were observed in the *Lin28a*[−/−] embryos that natively had only 12 pairs of ribs (*Figure 2—figure supplement 2A*). In the cervical region, the severity of skeletal patterning defects correlated with the genotype of *Lin28a* (*Figure 2—figure supplement 2B*–2F). After RA treatment, some *Lin28a*[+/−] embryos exhibited the C1/C2 fusion phenotype (*Figure 2—figure supplement 2D*), whereas *Lin28a*[−/−] embryos showed more severe defects that were characterized by fusion of the exoccipital bone with C1 and C2 (*Figure 2—figure supplement 2E and F*). These results show that perturbation of Hox genes by RA administration enhances the *Lin28a*[+/-] and [-/-] phenotypes. In particular, since RA administration in *Lin28a*[+/-] mice results in the same phenotype as untreated *Lin28a*[-/-], it is possible that dysregulation of *Hox* genes is responsible for the skeletal patterning defects in *Lin28a*[−/−] mice (*Figure 2E*).

## *Lin28a* regulates *Cbx2* expression via *let-7* repression

We examined the molecular mechanism underlying the *Lin28a*-mediated regulation of *Hox* gene expression during embryogenesis. Since *Lin28a* is known as a negative regulator of *let-7* biogenesis by interfering with Drosha processing of pri-let-7 (*Newman et al., 2008*; *Viswanathan et al., 2008*), and by TUT4-mediated terminal uridylation and inhibition of Dicer processing (*Heo et al., 2009*; *Chang et al., 2013*), we examined the microRNA expression profile of *Lin28a*[−/−] mice. TaqMan microRNA array analyses were performed on E9.5 embryos. Consistent with previous reports (*Viswanathan et al., 2008*; *Rybak et al., 2008*), we found that mature microRNAs of *let-7*-family members were significantly accumulated in *Lin28a*[−/−] mice (*Figure 3A*). These results were also confirmed by q-PCR analysis of *let-7*-family members (*Figure 3B*). There was no difference between Wt and *Lin28a*[−/−] mice with regards to the expression of either *mir-10* and *mir-196* family microRNAs, which are regulators of the spatial expression of *Hox* genes and of vertebral specification (*Woltering and Durston, 2008*; *Yekta et al., 2004*; *Hornstein et al., 2005*; *Figure 3C*). Consistent with previous reports (*Heo et al., 2009*; *Chang et al., 2013*), these results imply that the ablation of Lin28a promotes the specific accumulation of *let-7* family microRNAs during embryogenesis.

We then sought a potential target gene for *let-7*, which may be involved in the skeletal transformations observed in *Lin28a*[−/−] mice. We performed comprehensive screening for the *let-7* target candidate genes using the following criteria; 1) *let-7* target genes, computationally predicted using TargetScan (856 genes), and 2) annotated genes responsible for posterior transformations and of which knockout mice show vertebrae that are similar to those observed in *Lin28a*[-/-] mice, as screened by Mouse Genome Informatics (115 genes). We found that five of the genes (*Arl4d*, *Cbx2*, *Cbx5*, *Dusp4*, and *E2f6*) satisfied both criteria (*Figure 3D*). *Arl4d* and *E2f6* have been identified as potential *let-7* target genes (*Johnson et al., 2007*; *Li et al., 2015*), suggesting that this screening successfully extracted candidate genes. Three of the five genes (*Cbx2*, *Cbx5*, and *E2f6*) are PcG genes or *Polycomb*-associated genes (*Core et al., 1997*; *Nielsen et al., 2001*; *Courel et al., 2008*), suggesting that *Cbx5* and *E2f6*, as well as *Cbx2*, might be involved in *Hox* gene dysregulation via histone modifications and chromatin structural changes in *Lin28a*[-/-] mice. Based on this screening, we examined if those five genes are true targets of *let-7* by Luciferase assay. We generated reporter constructs of luciferase-*let-7* target site-mutated 3'UTR sequence of each gene, and quantified *let-7*-dependent reporter activity in comparison with a Luciferase-wild type 3'UTR sequence construct (*Figure 3E*). We found that three of the five genes, *Cbx2*, *Cbx5,* and *E2f6*, were down-regulated in a *let-7*-dependent manner, whereas this down-regulation effect was not observed in the *let-7* target site mutated

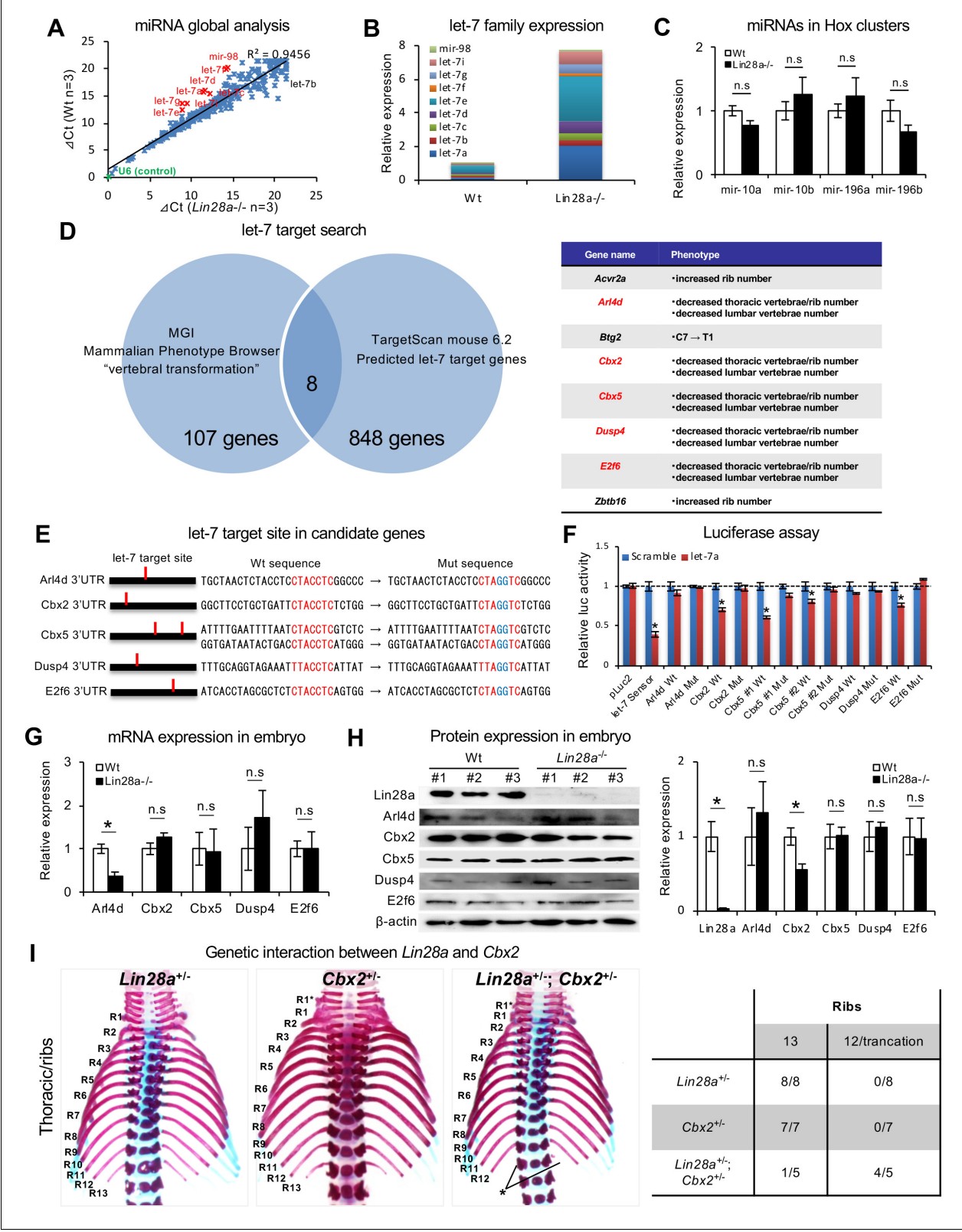

**Figure 3.** *Let-7* targets the polycomb gene directly. (**A**) Comparison of microRNA expression in Wt and *Lin28a*[−/−] embryos at E9.5. (**B, C**) q-PCR analyses of *let-7*-family members (**B**) and *Hox*-embedded microRNAs (**C**). In (**B**), data are expressed as the mean (n = 3), and the relative amount of total *let-7* microRNAs is shown. (**D**) *let-7* target search with TargetScan and Phenotype Browser. (**E**) The *let-7* target site in the 3'UTR sequence of candidate genes. The *let-7* seed-matched sequence and mutated sequence are shown in red and blue, respectively. (**F**) Luciferase reporter activity in the

*Figure 3 continued on next page*

*Figure 3 continued*

presence/absence of the *let-7* target site in 3'UTR sequence. (**G–H**) qPCR and western blot analyses of candidate genes. (**I**) Dorsal views of thoracic vertebrae and ribs. Single heterozygous mutants (left and middle panels) and a double heterozygous mutant (right panel) are shown. R1–R13, 1st to 13th ribs; asterisk, the ablation or truncation of the 13th rib. See also *Figure 3—figure supplement 1*. (**J**) Frequency of rib defects in mutant mice. All data are expressed as the mean ± SEM (n = 3). *p<0.05. n.s., not significant.

The online version of this article includes the following source data and figure supplement(s) for figure 3:

**Source data 1.** Source data related to panel A-C, and F-H.
**Figure supplement 1.** Skeletal defects in *Cbx2* mutant mice.
**Figure supplement 2.** Expression level of Cbx2 in *Lin28a* mutant embryo.

construct. In contrast, the potential let-7 target sequence of *Arl4d* and *Dusp4* did not affect the expression of luciferase. While these results in HEK293T cells with partial 3'UTR sequences do not completely exclude the possibility that *Dusp4* and *Arl4d* are not target genes of let-7, they do suggest that *Cbx2*, *Cbx5* and *E2f6* are direct targets of *let-7* (*Figure 3F*).

To confirm that these genes are affected by the *Lin28/let-7* axis in vivo, we performed mRNA and protein expression analyses on somite and neural tubes. qPCR analyses showed that *Arl4d* was significantly downregulated in *Lin28a*$^{-/-}$ embryos (*Figure 3G*). However, the luciferase assay revealed that luciferase expression was not affected by a let-7 target site mutation in the Ard4 3' UTR sequence, suggesting that *Arl4d* is not a direct target of *let-7*. Protein expression analyses revealed that *Cbx2* was the only gene that was significantly downregulated in *Lin28a*$^{-/-}$ embryos, and also its expression was affected in a *let-7*-dependent manner (*Figure 3F and H*). These findings suggest that *Cbx2* is, at least in part, a molecular target of the *Lin28a/let-7* pathway in skeletal patterning.

*Cbx2* is one of the PcG genes that regulates *Hox* genes via histone modification, and ablation of Cbx2 shows skeletal patterning defects in mice (*Core et al., 1997*; *Nielsen et al., 2001*; *Courel et al., 2008*). We considered that decreased expression of Cbx2 might cause the abnormal skeletal formation found in *Lin28a*$^{-/-}$ mice. Therefore, we examined whether decreasing the expression level of Cbx2 in Lin28a+/-could induce a similar phenotype as *Lin28a*$^{-/-}$ mice. We generated *Cbx2* mutant mice using CRISPR/Cas9 and interbred the *Cbx2* mutant with *Lin28a*$^{+/-}$ mice (*Figure 3—figure supplement 1A*). The *Cbx2* homozygous mutants exhibited skeletal patterning defects (*Figure 3—figure supplement 1B*–1E): fusion of C1 and C2 vertebrae, additional rib formation from the 7th cervical vertebra, T1 to T2 transformation of the spinous process, and 13th rib truncation. Similar observations were reported for Cbx2-null mice (*Core et al., 1997*; *Katoh-Fukui et al., 1998*). We generated double heterozygous mutants of Lin28a and Cbx2 (*Lin28a*$^{+/-}$; *Cbx2*$^{+/-}$) and analyzed their skeletal patterning. The double heterozygous mice showed ablation or truncation of the 13th pair of ribs, although the *Lin28*$^{+/-}$ and *Cbx2*$^{+/-}$ single mutants did not show any obvious phenotypic irregularities (*Figure 3I and J*). These results show that decreased expression of Cbx2 enhances the *Lin28a*$^{+/-}$ phenotypes. In particular, since *Lin28a*$^{+/-}$;Cbx2$^{+/-}$ double mutant mice showed the deformation of the 13th pair of ribs, which was similar to the phenotype observed in *Lin28a*$^{-/-}$ mice, it was possible that dysregulation of *Cbx2* is responsible for the phenotype of *Lin28a*$^{-/-}$ mice. Together, these results indicate that *let-7* directly regulates *Cbx2,* and that genetic interactions exist between *Lin28a* and *Cbx2*. Furthermore, they suggest that *Lin28a/let-7* reciprocal feedback regulates Cbx2 expression, and that this pathway contributes to the regulation of proper skeletal patterning during embryogenesis.

## The *Lin28a/let-7* pathway modulates PRC1 occupancy at posterior *Hox* loci

*Hox* gene expression is epigenetically restricted to unique spatiotemporal patterns during embryogenesis by PcG genes (*Soshnikova and Duboule, 2009*). To determine if the *Lin28a/let-7/Cbx2* axis regulates skeletal patterning via *Hox* gene expression, we analyzed histone modifications and PcG occupancy at the *Hox* loci. We performed chromatin immunoprecipitation (ChIP) and q-PCR analyses on E9.5 somites and neural tubes (*Figure 4A*). For each assay, ChIP was performed on a pool of dissected somites and neural tubes from ten embryos (as n = 1). First of all, we analyzed the repressive histone modification (H3K27me3) at *Hoxa* cluster loci of Wt. We found the promoter regions of *Hoxa3*, *Hoxa9*, *Hoxa10*, *Hoxa11*, and *Hoxa13* exhibited a high concentration of histone H3K27me3

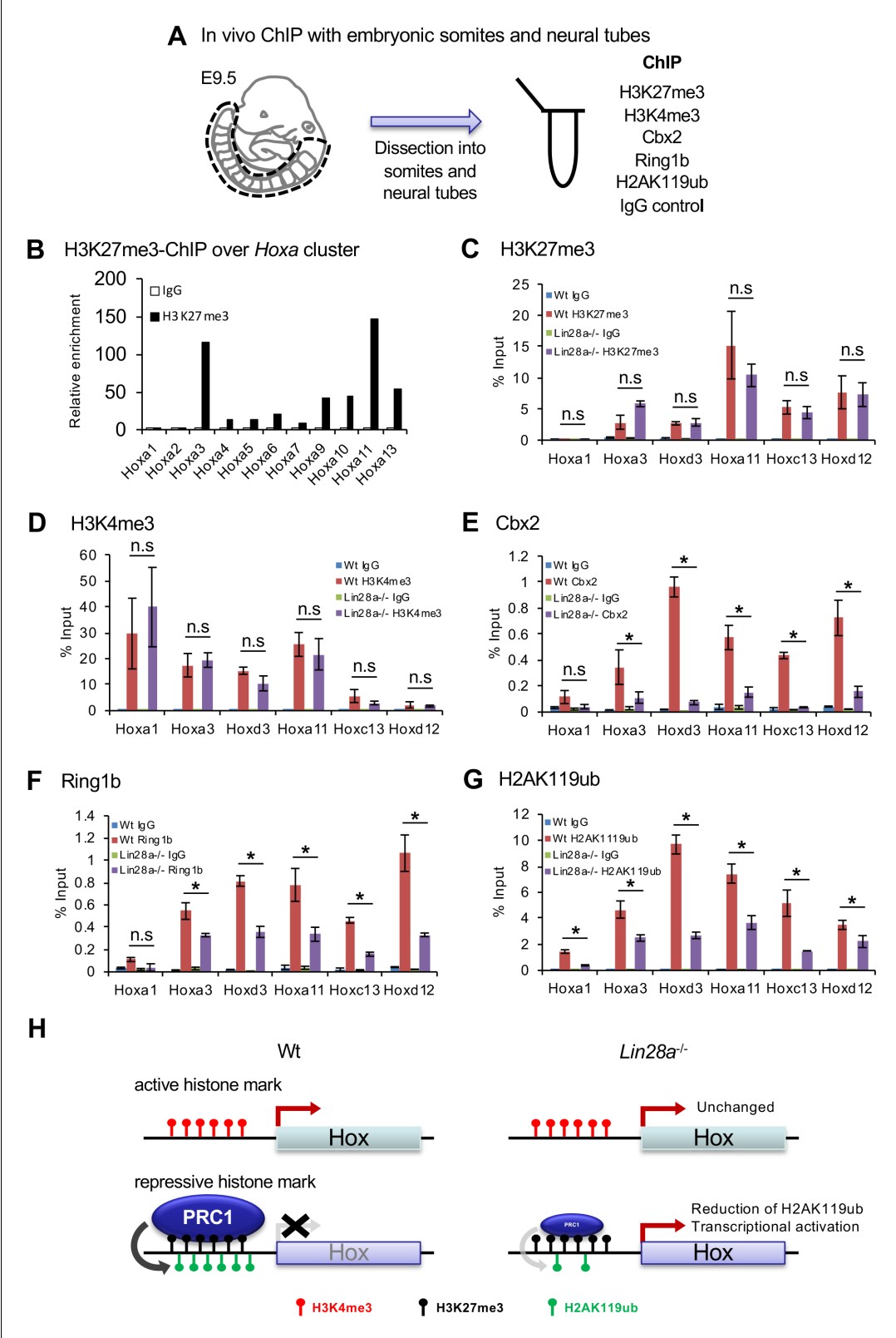

**Figure 4.** Histone modifications and polycomb occupancy at *Hox* loci in *Lin28a*[-/-] mice. (**A**) Schematic diagram of the experimental procedure for ChIP analysis. (**B**) ChIP and q-PCR analyses of H2K27me3 in *Hox* A cluster genes in Wt embryos. (**C–G**) ChIP and q-PCR analyses of H3K27me3 (**C**), H3K4me3 (**D**), Cbx2 (**E**), Ring1b (**F**), and H2AK119ub (**G**). Percentages of immunoprecipitated DNA compared with the input are shown. (**H**) Summary of the chromatin state of *Hox* loci in Wt and *Lin28a*[−/−] embryos. All data are expressed as the mean ± SEM (n = 3). *p<0.05. n.s., not significant.

*Figure 4 continued on next page*

*Figure 4 continued*

The online version of this article includes the following source data for figure 4:

**Source data 1.** Source data related to panel B-G.

(*Figure 4B*). Intriguingly, the same genes were upregulated in $Lin28a^{-/-}$ mice (*Figure 2A*), suggesting that these genes are tightly regulated epigenetically and that the loss of repressive histone modifications leads to the upregulation of these *Hox* genes.

Based on the analysis of phenotype of the $Lin28a^{-/-}$ skeletal transformation (*Figure 1B–I*) and *Hox* cluster gene expression pattern of $Lin28a^{-/-}$ embryos (*Figure 2A–C*), we focused on *Hoxa3* and *Hoxd3*, which are involved in C1-C2 malformation and partial fusion in knockout mice (*Condie and Capecchi, 1994*). In addition, we focused on *Hoxa11*, which has been reported as the responsible gene for T13 to L1 skeletal transformation in mutated mice (*Small and Potter, 1993*), *Hoxd12* and *Hoxc13*, which were upregulated in $Lin28a^{-/-}$ embryos, and *Hoxa1* as a representative of the anterior *Hox* genes (*Figure 4C–G*). Subsequently, we performed ChIP and q-PCR analyses using anti-H3K27me3 and anti-H3K4me3 antibodies in Wt and $Lin28a^{-/-}$ embryos (*Figure 4C and D*). We found that for histone H3 modifications, both K27me4 and K4me3 were not altered in $Lin28a^{-/-}$ embryos compared with Wt (*Figure 4C and D*). We also performed ChIP using antibodies against PRC1 components to test their occupancy at *Hox* loci (*Figure 4E and F*). Consistent with the expression level of Cbx2 (*Figure 3G*), we found at least a two-fold reduction of its binding at posterior *Hox* regions in $Lin28a^{-/-}$ mice (*Figure 4E*). Intriguingly, the occupancy of Ring1b, another component of PRC1 (*Suzuki et al., 2002*), and H2AK119 ubiquitination (H2AK119ub) which is catalyzed by Ring1b (*Suzuki et al., 2002*), were also reduced in $Lin28a^{-/-}$ mice (*Figure 4F–G*). Because each posterior *Hox* gene (*Hoxa11*, *Hoxc13*, and *Hoxd12*) is located on distinct chromosomes, these results indicate a critical role for the *Lin28a*/*let-7* axis in PcG-mediated *Hox* gene repression. Taken together, these findings suggest that Cbx2 repression by *let-7* leads to the reduction of PRC1 occupancy at the *Hox* loci and the transcriptional initiation of posterior *Hox* genes (*Figure 4H*).

### *Let-7* knockdown rescues *Hox* gene dysregulation in $Lin28a^{-/-}$ cells

To further elucidate the importance of the direct regulation of *let-7* by Lin28a during *Hox* gene regulation, we tested whether *Hox* gene dysregulation could be rescued by knockdown of *let-7*-family microRNAs. To accomplish this, $Lin28a^{-/-}$ embryonic stem (ES)-like cells were established from mutant blastocysts. Each $Lin28a^{-/-}$ clone resembled Wt cells (*Figure 5A*), and we confirmed that the Lin28a protein was not detected in $Lin28a^{-/-}$ ES cells (*Figure 5B*). These colonies showed high alkaline phosphatase activity (*Figure 5A*) and also expressed pluripotent factors (*Figure 5C*). As observed in $Lin28a^{-/-}$ embryos (*Figure 3B*), global accumulation of *let-7*-family microRNAs was observed in the mutant cells (*Figure 5D*).

In the following experiments, we differentiated ES cells to embryoid bodies. ES cells and embryoid bodies require different PRC1 components to maintain their state. ES cells are maintained in an undifferentiated state, using Cbx7 containing PRC1. On the other hand, when ES cells exit the pluripotent state and differentiate into embryoid bodies, Cbx2 is expressed and becomes a component of PRC1 (*Morey et al., 2012*). Thus, we utilized embryoid bodies as an appropriate model to analyze *Hox* genes via *Lin28*/*Let-7*/*Cbx2* axis. Embryoid bodies were produced from each clone and expression changes of *Hox* genes were analyzed. *Hox* genes were upregulated upon differentiation in these embryoid bodies, suggesting that a recapitulation of the *Hox* gene upregulation observed in $Lin28a^{-/-}$ mice occurred in $Lin28a^{-/-}$ ES-like cells (*Figure 5E*).

Next, we knocked down the *let-7* family in $Lin28a^{-/-}$ ES-like cells using the CRISPR/Cas9 system to test if *Hox* gene upregulation could be rescued by the reduction of *let-7* microRNAs. The major *let-7* family is composed of 11 genes (*a-1*, *a-2*, *b*, *c-1*, *c-2*, *d*, *e*, *f-1*, *f-2*, *g*, and *i*), and we performed the knockdown of this series of *let-7* genes using guide RNAs targeting *let-7s* (*Figure 5F*). The clone that yielded a highly efficient deletion of *let-7* microRNAs in $Lin28a^{-/-}$ cells ($Lin28a^{-/-}$; *let-7*KD) was selected for further analyses. We confirmed the accumulation of *let-7* in $Lin28a^{-/-}$ cells and the drastic reduction in $Lin28a^{-/-}$; *let-7*KD clones (*Figure 5G*). qPCR analysis revealed that dysregulation of Hoxa11 and Hoxd12 was rescued in $Lin28a^{-/-}$; *let-7*KD clones (*Figure 5H*). Moreover, we also

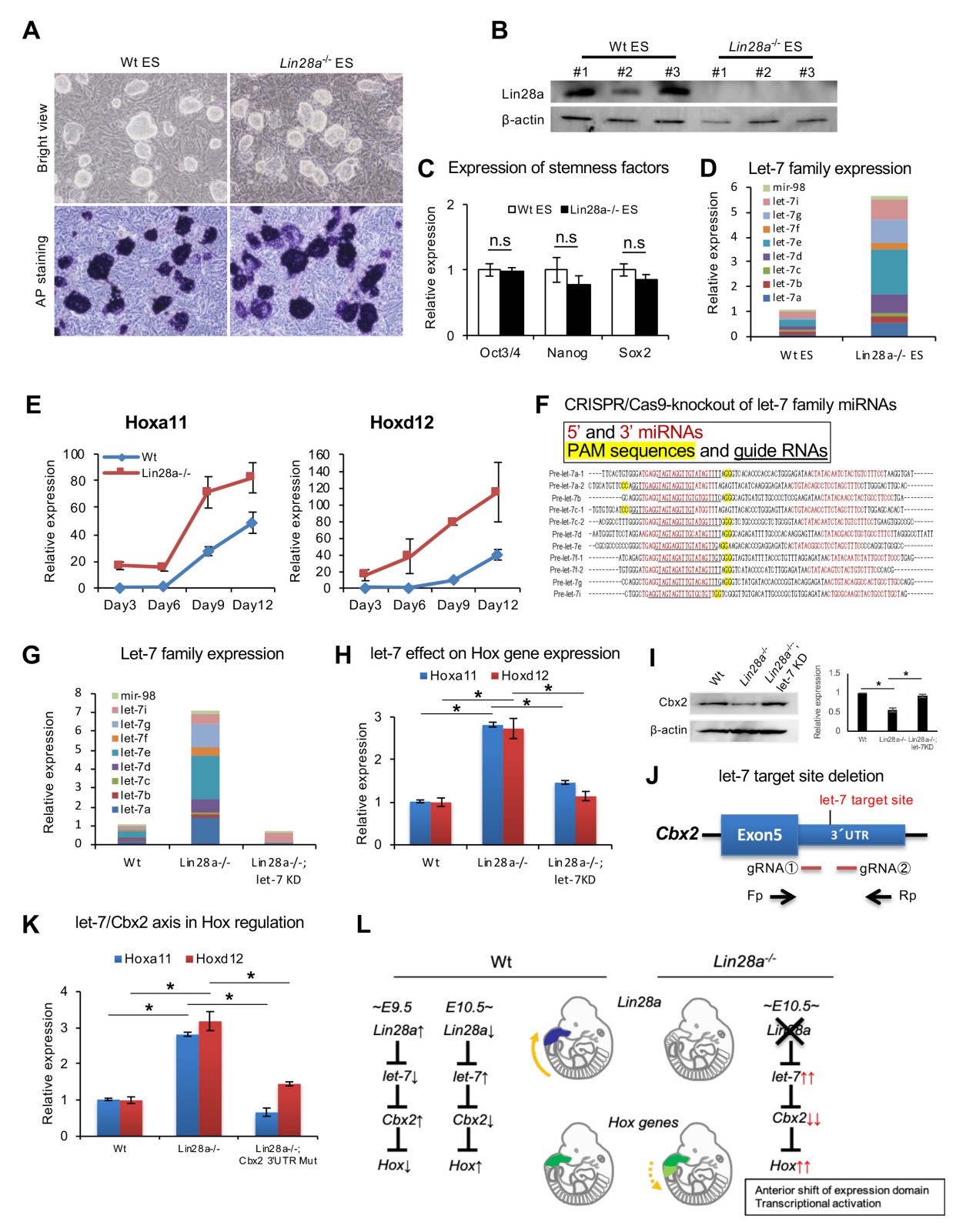

**Figure 5.** Knockdown of *let-7* can reverse *Hox* gene dysregulation. (**A**) Morphology (top panels) and alkaline phosphatase activity (bottom panels) of Wt and *Lin28a*⁻/⁻ ES-like cells. (**B**) Western blot analysis of Lin28a in ES-like cells. β-actin is shown as a loading control. (**C**) q-PCR analysis of stemness factors. (**D**) q-PCR analysis of *let-7*-family members. The level of expression relative to total let-7 amount in Wt is shown. (**E**) q-PCR analyses of *Hoxa11* and *Hoxd12* over a time course of 3, 6, 9, 12 days following embryoid body formation. (**F**) Precursor sequences of *let-7*-family members and guide RNAs

*Figure 5 continued on next page*

*Figure 5 continued*

for *let-7* targeting Let-7 mature microRNAs are shown in red. The protospacer adjacent motif (PAM) sequence for hCas9 is highlighted in yellow, and targeting sequences are underlined. (G) *Let-7* expression in Wt, *Lin28a*$^{-/-}$ and *Lin28a*$^{-/-}$; *let-7* KD cells. The level of expression relative to total let-7 amount in Wt is shown. (H) *Let-7* knockdown rescues *Hox* gene dysregulation in *Lin28a*$^{-/-}$ cells. (I) Cbx2 expression level of Wt, *Lin28a*$^{-/-}$ and *Lin28a*$^{-/-}$; *let-7* KD derived EBs. β-actin is shown as a loading control. (J) Schematic diagram of *let-7* target site deletion from *Cbx2* 3'UTR and genotyping via PCR of mutant clones. (K) q-PCR analyses of *Hoxa11* and *Hoxd12* following embryoid body formation. (L) Schematic diagram of *Lin28a*/*let-7* mediated *Hox* gene regulation. All data are expressed as the mean ± SEM (n = 3). n.s., not significant.

The online version of this article includes the following source data for figure 5:

**Source data 1.** Source data related to panel C-E, G-H, and K.

confirmed that a decreasing Cbx2 protein expression level in embryoid bodies-derived *Lin28a*$^{-/-}$ ES-like cells was rescued by knock down of *let-7* (*Figure 5I*).

To directly prove that the *Lin28a*$^{-/-}$ phenotype results from the *let-7*-mediated down-regulation of *Cbx2*, we established *Lin28a*$^{-/-}$ ES cells with *let-7* target site deletion from *Cbx2* 3'UTR (*Lin28a*$^{-/-}$; *Cbx2* 3'UTR mutant) using CRISPR/Cas9 system, and examined whether *let-7* target site deletion from *Cbx2* 3'UTR could rescue *Hox* gene dysregulation. Two guide RNAs targeting the *let-7* binding site in *Cbx2* 3'UTR were constructed and transfected with the Cas9 expression vector into *Lin28a*$^{-/-}$ ES-like cells for the establishment of *Cbx2* 3'UTR mutant cell lines (*Figure 5J*). Furthermore, we generated embryoid bodies from Wt, *Lin28a*$^{-/-}$, and *Lin28a*$^{-/-}$; *Cbx2* 3'UTR mutant clones in the same manner as that of the *let-7* knock down experiment (*Figure 5G–I*), and the expression level of *Hox* genes were analyzed. Consistent with the result of *let-7* knock down (*Figure 5G–I*), we found that *Hoxa11* and *Hoxd12* were up-regulated in *Lin28a*$^{-/-}$ cells, and that this abnormal expression was absent in *Lin28a*$^{-/-}$; *Cbx2* 3'UTR mutant cells (*Figure 5K*). These results suggest that *let-7*-mediated Cbx2 repression is, at least in part, responsible for *Hox* gene dysregulation in Lin28a$^{-/-}$ mice. Taken together, our results suggest that the upregulation of *let-7* leads to decreased PRC1 occupancy, which causes the disruption of the 'Hox code,' thus indicating the potential role of the *Lin28a*/*let-7* pathway in skeletal patterning via *Polycomb*-mediated *Hox* gene regulation (*Figure 5L*).

## Discussion

The body plan along the anteroposterior axis is tightly regulated by *Hox* genes. During development, each *Hox* gene must be activated at a precise position with precise timing. Spatiotemporal regulation via chromatin conformational changes is essential for *Hox* gene expression and for subsequent anteroposterior patterning (*Soshnikova, 2014*); however, the molecular mechanisms behind these processes are not fully understood. In this study, we demonstrated the fundamental role of the *Lin28a*/*let-7* pathway in skeletal patterning and vertebral specification. Lin28a-mediated repression of *let-7* biogenesis is required for *Cbx2* expression and *Hox* gene repression by PcG genes. It is known that the deletion mutants of the *Hox* early enhancer exhibit anterior transformations of vertebrae because of the heterochrony of *Hox* gene expression (*Juan and Ruddle, 2003*). In our *Lin28a*$^{-/-}$ mice, posterior transformations were observed in the thoracic region (*Figure 1D–I*), suggesting that developmental timing of *Hox* gene initiation occurs earlier than in Wt mice. Consistent with this speculation, precocious expression of *Hoxc13* causes premature arrest of axial extension, similar to that of *Lin28a*$^{-/-}$ mice (*Young et al., 2009*; *Mallo et al., 2010*). This indicates that the tail truncation observed in *Lin28a*$^{-/-}$ mice might be caused by spatiotemporal dysregulation of *Hoxc13* (*Figure 2B and C*). Recently, two independent groups reported the function of the Lin28 family as regulators of trunk elongation (*Aires et al., 2019*; *Robinton et al., 2019*). Tail bud specific overexpression of the *Lin28* family increased caudal vertebrae number (*Aires et al., 2019*; *Robinton et al., 2019*). Moreover, the loss of *Lin28* in the tail bud resulted in the reduction of caudal vertebrae number (*Robinton et al., 2019*). These results are consistent with our *Lin28a*$^{-/-}$ mice phenotypes with short stature and shortened tails (*Figure 1B* and *Figure 1—figure supplement 2*). Furthermore, *Aires et al., 2019* showed that Lin28 and Hox13 had opposite functions in tail bud proliferation,

suggesting that the balance of the expression of those two genes, which might be regulated by GDF signaling, is one of the determinants of tail length. Our results revealed the epigenetic inhibition of HoxPG13 by the *Lin28a/let-7/Cbx2* pathway, which might be one of the mechanisms that explains the antagonistic function of Lin28a and HoxPG13 in axial elongation as well as in skeletal patterning.

In contrast to the short tailed-phenotype caused by HoxPG13 inhibition, the molecular mechanisms underlying the other skeletal patterning defects found in the cervical, thoracic, and lumbosacral regions were still unknown. Aires et al. and Robinton et al. showed that Lin28a regulated the cell fate choice between mesodermal cells and neural cells; however, no skeletal transformations were observed in their *Lin28a* mutant mice (*Aires et al., 2019*; *Robinton et al., 2019*). Further analyses are required to determine if the skeletal patterning defects found in the cervical, thoracic, and lumbosacral regions of *Lin28a*$^{-/-}$ mice are caused by the dysregulation of cell fate choice. Based on the analysis of phenotype of the *Lin28a*$^{-/-}$ skeletal transformations (*Figure 1B–I*), we focused on *Hoxa3/d3* for the C1-C2 malformation and partial fusion (*Condie and Capecchi, 1994*), *Hoxb8/c8* for rib patterning (*van den Akker et al., 2001*), and *Hoxa11* for T13 to L1 skeletal transformation (*Small and Potter, 1993*). However, these *Hox* genes showed no obvious difference in their expression patterns, although the expression of the genes was up-regulated (*Figure 2A* and *Figure 2—figure supplement 1*). Skeletal transformations are mainly caused by the altered expression pattern of *Hox* genes. However, it is also possible that changes in the expression amount of Hox genes may be involved in skeletal patterning. For instance, *Dll1* enhancer-driven *Hoxb6* transgenic mice show ectopic rib-like structures in the cervical, lumber, sacral and caudal regions. However, malformation of the axial skeleton was shown even in the thorax, which is the regular expressing region of *Hoxb6* (*Vinagre et al., 2010*). These results suggest that the *Hox*-code of this specific region might have been edited due to the elevated expression of a specific *Hox* gene, which might cause the morphological change of vertebrae in our *Lin28a*$^{-/-}$ mice.

PcG genes are regulators of the '*Hox* code' at the level of chromatin structure, which occurs via epigenetic histone modifications (*Mallo and Alonso, 2013*; *Soshnikova, 2014*). In ES cells, *Hox* genes are silenced in a bivalent state containing both H3K27me3, a repressive, and H3K4me3, an active histone marker. During development, the epigenetic status of *Hox* loci is dynamically balanced by PcG genes and *Trithorax* group (TrxG) genes, which are required for the trimethylation of H3K4. PcG genes should be repressed prior to the initiation of *Hox* gene expression to open the chromatin along the anteroposterior axis. However, the precise molecular mechanisms underlying the inhibition of the expression of PcG genes during embryogenesis are not fully understood. Here, we provide evidence that the *Lin28a/let-7* pathway is, at least in part, one of the mechanisms involved in the regulation of PcG genes (*Figure 3*). Cbx2 is required for the binding of PRC1 to target loci and recognition of H3K27me3, and these processes are catalyzed by Ezh2, the main component of PRC2. *Ezh2* is directly targeted by *let-7* microRNAs in primary fibroblasts and cancer cells (*Kong et al., 2012*). In contrast with those findings, there were no apparent differences in the level of H3K27me3 at *Hox* loci in *Lin28a*$^{-/-}$ mice (*Figure 4C*). *Ezh2*$^{-/-}$ embryos died at the peri- and post-implantation stages (*O'Carroll et al., 2001*), whereas mutant mice of the PRC1 genes exhibited skeletal transformations (*van der Lugt et al., 1994*; *Akasaka et al., 1996*; *Core et al., 1997*; *Suzuki et al., 2002*; *Li et al., 2011*; *Katoh-Fukui et al., 1998*) that were similar to those of *Lin28a*$^{-/-}$ mice (*Figure 1D–I*). These observations suggest that the *Lin28a/let-7* pathway is involved in the later phases of epigenetic silencing of *Hox* genes during skeletal patterning. Since *Lin28a*$^{+/-}$;*Cbx2*$^{+/-}$ double mutant mice showed the deformation of the 13$^{th}$ pair of ribs, which was similar to the phenotype observed in *Lin28a*$^{-/-}$ mice, it was possible that dysregulation of *Cbx2* is responsible for the phenotype of *Lin28a*$^{-/-}$ mice (*Figure 3I*). Moreover, we observed the reduction of PRC1 occupancy at *Hox* loci in *Lin28a*$^{-/-}$ mice (*Figure 4E and F*). These findings indicate that *let-7*-mediated Cbx2 repression leads to the reduction of PRC1 occupancy at *Hox* loci, resulting in the transcriptional initiation of posterior *Hox* genes (*Figure 4H*).

In addition to epigenetic regulation by PcG genes, posttranscriptional regulation by microRNAs is also required for anteroposterior patterning. During mouse embryogenesis, mesoderm-specific ablation of Dicer, which is an RNase III enzyme that is required for microRNA biogenesis, results in a posterior shift in hindlimb position (*Zhang et al., 2011*), suggesting the involvement of microRNAs in normal skeletal patterning and vertebrae specification. Two microRNA families, *mir-10s* and *mir-196s*, are located in *Hox* clusters, and they are thought to regulate *Hox* gene expression and specify

the regional identities along the anteroposterior axis (*Heimberg and McGlinn, 2012*). It has also been reported that the *mir-17–92* cluster, which contains *mir-17*, *mir-18*, *mir-19*, *mir-20*, and *mir-92*, is required for normal skeletal patterning (*Han et al., 2015*). Although *Lin28a* is a regulator of let-7 microRNA biogenesis, the expression of these microRNAs was not altered in the *Lin28a*$^{-/-}$ mice compared with Wt animals (*Figure 3A and C*). These results suggest that the *Lin28a/let-7* pathway acts independently of these microRNAs in *Hox* gene regulation. *Mir-10s* and *mir-196s* are involved in the spatial regulation of *Hox* genes to shut down target *Hox* genes in specific regions (*Heimberg and McGlinn, 2012*), whereas *let-7* might be required for temporal activation of Hox genes via Lin28a downregulation during development. These results suggest that *let-7* can be distinguished from other microRNAs in skeletal patterning, and that the *Lin28a/let-7* pathway links posttranscriptional regulation to PcG-mediated epigenetic regulation in *Hox* gene regulation.

MicroRNAs are thought to regulate hundreds of target genes and to modulate multiple biological processes, and hence, the accumulation of *let-7* observed in *Lin28a*$^{-/-}$ mice might lead to extensive disorders of gene regulatory networks. It is well known that the *let-7* family regulates *c-Myc*, *K-ras*, *Hmga2*, and other genes that are involved in cell proliferation and oncogenesis (*Mayr et al., 2007*; *Lee and Dutta, 2007*; *Johnson et al., 2005*; *Sampson et al., 2007*). Knockout mice for these genes exhibit dwarfism caused by a reduction of cell proliferation that is similar to that observed in *Lin28a*$^{-/-}$ mice (*Zhou et al., 1995*; *Koera et al., 1997*; *Johnson et al., 1997*; *Trumpp et al., 2001*). These observations suggest that the growth defects and postnatal mortality of *Lin28a*$^{-/-}$ mice (*Figure 1—figure supplement 2*) may be attributed to the dysregulation of such genes; however, their requirement for skeletal patterning has not been characterized. Despite the contribution of these genes to the *Lin28a*$^{-/-}$ phenotype, it is noteworthy that there was a genetic interaction between *Lin28a* and *Cbx2* during skeletal patterning (*Figure 3I*). These results suggest that the *Lin28a/let-7/Cbx2* pathway is, at least in part, responsible for normal skeletal patterning. In addition to *Lin28a*, *Lin28b* regulates *let-7* biogenesis, and it is known that single nucleotide polymorphisms (SNPs) of the human *LIN28B* locus correlated with height and the timing of menarche (*Lettre et al., 2008*; *Perry et al., 2009*; *He et al., 2009*; *Ong et al., 2009*; *Widén et al., 2010*; *Sulem et al., 2009*). These studies suggest that the regulation of developmental timing by *Lin28b* is also conserved in mammals; however, its requirement in skeletal patterning is still unclear.

The expression level of Cbx2 was also downregulated in heterozygous *Lin28a*$^{+/-}$ (*Figure 3—figure supplement 2*). This indicates that Lin28a expression in heterozygous *Lin28a*$^{+/-}$ is reduced to less than half (*Figure 3—figure supplement 1C*), suggesting that other target molecules regulated by Lin28a might be involved in this skeletal transformation phenotype. In addition to the regulation of *let-7*, it is also known that Lin28a and its homolog, Lin28b, bind to and modulate the translation efficiency of specific mRNAs, such as *Igf2*, *Oct4*, *Ccnb1*, *Cdk6*, *Hist1h2a*, and *Bmp4* (*Xu et al., 2009*; *Ma et al., 2013*; *Qiu et al., 2010*; *Xu and Huang, 2009*). Moreover, recent HITS-CLIP and PAR-CLIP technology identified a variety of mRNAs as *Lin28* family targets (*Wilbert et al., 2012*; *Madison et al., 2013*; *Hafner et al., 2013*; *Cho et al., 2012*). Among them, two studies showed that the *Lin28* family might have the potential to bind specific *Hox* genes in HEK293T, DLD1, and Lovo cell lines (Lin28a to *Hoxa9*, *a11*, *b4*, *b6*, *b9*, *c4*, *d11*; Lin28b to *Hoxa9*, *b3*, *b4*, *b7*, *b8*, *b9*, *d13*) (*Hafner et al., 2013*; *Madison et al., 2013*), although CLIP-Seq analysis with ES cells did not show that (*Cho et al., 2012*). Moreover, *Cbx5* is a *Lin28a* target gene as well as one of the potential *let-7* targets. *Cbx5* encodes a heterochromatin binding protein, and the depletion of this gene causes skeletal defects in mice, although the protein level of Cbx5 was not altered in *Lin28a*$^{-/-}$ mice. These previous reports and our results imply that both *let-7*-dependent and -independent function of *Lin28a* might affect skeletal patterning during development. However, further studies are required to deepen the understanding of the developmental functions of *Lin28* family and its involvement in skeletal patterning.

Taken together, our results suggest that the negative feedback between *Lin28a* and *let-7* regulates the PRC1 component, *Cbx2*, and the subsequent spatiotemporal expression of *Hox* genes during mammalian embryogenesis. The loss of Lin28a caused skeletal transformations via the premature loss of PRC1 at the promoter region of posterior *Hox* genes, thus establishing a new role of the *Lin28a/let-7* pathway in the modulation of the 'Hox code.' It is of interest to test whether this role of *Lin28a/let-7* in *Hox* regulation was acquired in the evolutional process, or if it has always been involved in heterochrony in *C. elegans*.

# Materials and methods

## Key resources table

| Reagent type (species) or resource | Designation | Source or reference | Identifiers | Additional information |
|---|---|---|---|---|
| Antibody | anti-Arl4d | Santa Cruz | SC-271274 | mouse monoclonal antibody, for western blot, at 1:500 |
| Antibody | anti-b-actin | Sigma | A5316 | mouse monoclonal antibody, for western blot, at 1:2000 |
| Antibody | anti-Cbx2 | Abcam | ab80044 | Rabbit polyclonal antibody, for western blot, at 1:500 |
| Antibody | anti-CBX2 | Bethyl Laboratories | A302-524A | Rabbit polyclonal antibody, for ChIP |
| Antibody | anti-Cbx5 | Cell Signaling Technology | #2616S | Rabbit polyclonal antibody, for western blot, at 1:1000 |
| Antibody | anti-DIG-AP Fab fragment antibody | Roche | 1-093-274 | sheep polyclonal antibody, for in situ hybridization |
| Antibody | anti-Dusp4 (MKP-2) | Santa Cruz | SC-1200 | Rabbit polyclonal antibody, for western blot, at 1:250 |
| Antibody | anti-E2f6 | Santa Cruz | SC-8366 | goat polyclonal antibody, for western blot, at 1:500 |
| Antibody | anti-Lin28a | Cell Signaling Technology | #3978S | Rabbit polyclonal antibody, for western blot, at 1:1000 |
| Antibody | anti-mouse IgG HRP-conjugated | Sigma | A2304 | goat affinity isolated antibody, for western blot, at 1:2000 |
| Antibody | anti-rabbit IgG HRP-conjugated | Sigma | A6154 | goat affinity isolated antibody, for western blot, at 1:2000 |
| Antibody | anti-trimethyl-histone H3 (Lys27) | Millipore | #07–449 | Rabbit Polyclonal Antibody, for ChIP |
| Antibody | anti-trimethyl-histone H3 (Lys4) | Millipore | #07–473 | Rabbit Polyclonal Antibody, for ChIP |
| Antibody | normal rabbit IgG | Santa Cruz | SC-2027 | Rabbit Polyclonal Antibody, for ChIP |
| Antibody | RING1B (D22F2) XP rabbit monoclonal antibody (mAb) | Cell Signaling Technology | #5694S | rabbit monoclonal antibody, for ChIP |
| Cell Lines | HEK293T cells | ATCC | RRID:CVCL_0063 | |
| Cell Lines | Wt or Lin28a-/-ES like cells | Materials and methods section | N/A | |
| Chemical compound, drug | 2-mercaptoethanol | Gibco | #21985023 | |
| Chemical compound, drug | acetic anhydride | Wako | #011–00276 | |

*Continued on next page*

*Continued*

| Reagent type (species) or resource | Designation | Source or reference | Identifiers | Additional information |
|---|---|---|---|---|
| Chemical compound, drug | Alcian Blue | Sigma | A5268-10G | |
| Chemical compound, drug | Alizarin Red S | Sigma | A5533-25G | |
| Chemical compound, drug | Chaps | Dojindo Molecular Technologies | 349–04722 | |
| Chemical compound, drug | CHIR 99021 | Wako | 034–23103 | |
| Chemical compound, drug | Fast Green FCF | Sigma | F7258-25G | |
| Chemical compound, drug | Fast Red Violet LB Salt | Sigma | F3381-5G | |
| Chemical compound, drug | formamide | Sigma | SIGF5786 | |
| Chemical compound, drug | G-418 Sulfate | Wako | 074–05963 | |
| Chemical compound, drug | glycine | Wako | #077–00735 | |
| Chemical compound, drug | heparin | Nacalai Tesque | 17513–96 | |
| Chemical compound, drug | NBT/BCIP | Roche | #1697471 | |
| Chemical compound, drug | PD0325901 | Wako | 162–25291 | |
| Chemical compound, drug | PFA | Wako | #162–16065 | |
| Chemical compound, drug | Retinoic acid (all-trans) | Wako | 182–01111 | |
| Chemical compound, drug | sodium pyruvate | Gibco | #11360070 | |
| Chemical compound, drug | triethanolamine | Wako | 142–05625 | |
| Commercial assay, kit | Chemi-Lumi One | Nacalai Tesque | #07880 | |
| Commercial assay, kit | DirectPCR Lysis reagent | Viagen Biotech | #102 T | |
| Commercial assay, kit | ExoSAP-IT Express PCR Cleanup Reagents | ThermoFisher scientific | #75001 | |
| Commercial assay, kit | FugeneHD | Promega | E2312 | |
| Commercial assay, kit | GoTaq Flexi DNA Polymerase | Promega | M8298 | |
| Commercial assay, kit | Lipofectamine 2000 | Invitrogen | #11668019 | |
| Commercial assay, kit | MegaClear Transcription Clean-Up Kit | Invitrogen | AM1908 | |
| Commercial assay, kit | mMESSAGE mMACHINE T7 Kit | Invitrogen | AM1344 | |
| Commercial assay, kit | SuperSignal West Femto Maximum Sensitivity Substrate | Thermo Fisher Scientific | #34095 | |

*Continued on next page*

*Continued*

| Reagent type (species) or resource | Designation | Source or reference | Identifiers | Additional information |
|---|---|---|---|---|
| Commercial assay, kit | SYBR Green PCR Master Mix | Applied Biosystems | #4309155 | |
| Commercial assay, kit | TaqMan MicroRNA Assays | Applied Biosystems | *let-7a (#000377), let-7b (#002619), let-7c (#000379), let-7d (#002283), let-7e (#002406), let-7f (#000382), let-7g (#002282), let-7i (#002221), mir-98 (#000577), mir-10a (#000387), mir-10b (#002218), mir-196a (#241070), mir-196b (#002215), RNU6B (#001093)* | |
| Commercial assay, kit | TaqMan Rodent Micro RNA Array A and B | Applied Biosystems | #4398979 | |
| Commercial assay, kit | TaqMan Rodent Micro RNA Array B | Applied Biosystems | #4398980 | |
| Commercial assay, kit | TaqMan Universal Master Mix II, no UNG | Applied Biosystems | #4440040 | |
| Commercial assay, kit | the TaqMan MicroRNA Reverse Transcription kit | Applied Biosystems | #4366597 | |
| Peptide, recombinant protein | ESGRO Recombinant Mouse LIF Protein | Merck Millipore | ESG1107 | |
| Peptide, recombinant protein | Proteinase K recombinant PCR Grade | Roche | 03-115-887-001 | |
| Strains | Cbx2 deficient mice | Materials and methods section | N/A | |
| Strains | Lin28a deficient mice | Materials and methods section | N/A | |
| Strains | Meox2 Cre | The Jackson Laboratory | N/A | |
| Other | Dulbecco's Modified Eagle's medium (DMEM) | Sigma | D5796 | |
| Other | Glutamax | Gibco | #35050061 | |
| Other | Immobilon | Millipore | WBKLS0100 | |
| Other | nonessential amino acids (NEAAs) | Gibco | #11140050 | |
| Other | sheep serum | Thermo Fisher Scientific | 535–81301 | |
| Other | skim milk | Wako | #190–12865 | |
| Other | tRNA | Roche | 109–495 | |

## Generation of mutant mice

All animal experiments were performed in accordance with protocols approved by the Institutional Animal Care and Use Committee of the National Research Institute for Child Health and Development (permit numbers: 2004–003, 2014–001). To accomplish the *Lin28a* knockout, the targeting vector was constructed to replace the endogenous *Lin28a* locus with the Venus gene and PGK-neo cassette by homologous recombination in ES cells. The 5' and 3' sequences flanking the endogenous *Lin28a* locus were amplified by PCR from a C57BL/6N genomic *bacterial artificial chromosome* (BAC) clone (BACPAC Resource Center). The primer sequences used for homology arm cloning were as follows: 5' homology arm forward primer (Fp) NotI, 5'–TTGCGGCCGCGGCTCCCTTGCCTGGTCCTCCTGCCGATTC–3'; 5' homology arm reverse primer (Rp) SalI, 5'–GCGTCGACGGTCGTCTGCTGAGCCCGTGGCCCCGGG–3'; 3' homology arm Fp ClaI, 5'–GGATCGATTCGAGCTTGCGATTCAGCGGGCACACCTTAGG–3'; and 3' homology arm Rp AscI, 5'–AAGGCGCGCCAGGGTCTGGCAGCTGAGGAAGTTCCCCTAA–3'. These homology arms were cloned into a vector that incorporated both a neomycin-resistance cassette for positive selection and a diphtheria toxin A (*DT-A*) gene for negative selection. The targeting vector was linearized and electroporated into TT2F ES

cells. Recombinant ES clones were isolated after culture in medium containing the G418 antibiotic and screened for proper integration by Southern blotting using the 5' probe, 3' probe, and neo cassette sequence. Two clones exhibited proper integration, as validated by genomic sequencing, and were chosen for microinjection into 8 cell stage embryos. The resulting chimeric offspring were crossed to C57BL/6N mice and germ-line transmission was confirmed by Southern blotting and PCR. The floxed PGK-neo cassette was removed by crossing with Meox2-Cre mice (The Jackson Laboratory). Genotyping of *Lin28a* mutant mice was performed by PCR analysis. Genomic DNA was isolated from mouse tail snips. Each tail snip was incubated at 50°C with DirectPCR Lysis reagent with Proteinase K for more than 6 hr, followed by heating at 80°C for 1 hr, to inactivate Proteinase K. The tail lysate (1 μL) was used as a PCR template. Genotyping PCR was carried out using GoTaq Flexi DNA Polymerase, according to the manufacturer's protocol. The primer sequences used for *Lin28a* genotyping PCR were as follows: *Lin28a* KO genotyping 1, 5'–TACAAGCCACTGGAACACCA–3'; *Lin28a* KO genotyping 2, 5'–GGGGGTTGGGTCATTGTCTTT–3'; and *Lin28a* KO genotyping 3, 5'–GTTCTGCTGGTAGTGGTCGG–3'.

For CRISPR/Cas9-mediated gene targeting via nonhomologous end joining (*Wang et al., 2013*; *Inui et al., 2015*), the guide RNA containing the target sequence of the *Cbx2* CDS (CTGAGCAGCGTGGGCGAGC) was synthesized in vitro using mMESSAGE mMACHINE T7 Kit and was purified using MegaClear Transcription Clean-Up kit, according to the manufacturer's instructions. A mixture containing 250 ng/μL of guide RNA and hCas9 mRNA was microinjected into the cytoplasm of a 1 cell stage embryo (C57BL/6N background). For genotyping, genomic DNA was isolated from mouse tail snips. Genotyping PCR was carried out using GoTaq Flexi DNA Polymerase, according to the manufacturer's protocol. The primer sequences used for *Cbx2* genotyping PCR were as follows: *Cbx2* CDS genotyping Fp, 5'–CCCTCTGGCCAAACAATAGCTTTCCGCAGGGACC–3'; and *Cbx2* CDS genotyping Rp, 5'–GCGCCACTTGACCAGGTACTCCAGCTTGCCCTGC–3'. The PCR products were treated with ExoSAP-IT and were then used as a template for direct sequencing. Sequence analysis of the *Cbx2* CDS locus was performed using F0 offspring, and mice that carried frameshift mutations were selected for further analysis.

## HEK293T culture

HEK293T cells were purchased from the American Type Culture Collection (ATCC). Cells were maintained in Dulbecco's Modified Eagle's medium (DMEM) supplemented with 10% FBS and antibiotics. There is no mycoplasma contamination in this cell line.

## Establishment of ES-Like cells

*Lin28a*$^{-/-}$ blastocysts were harvested and cultured on mouse embryonic fibroblasts (MEFs) in ES culture medium (15% FBS, 4.5 g/L of D-glucose, 1 × Glutamax, 1 mM sodium pyruvate, 1 × nonessential amino acids (NEAAs), 0.1 mM 2-mercaptoethanol, and 1 × 10$^4$ units/mL of LIF in DMEM) with 3 μM of CHIR 99021 and 1 μM of PD0325901. Each colony was isolated and expanded, followed by genotyping PCR. Wt and *Lin28a*$^{-/-}$ ES-like cells were stained with NBT/BCIP solution to test for alkaline phosphatase activity. There is no mycoplasma contamination in these cells. Western blotting and q-PCR analyses were performed for each genotype, as described below.

## Western blotting

Whole-protein extracts from the somites and neural tubes of E9.5 embryos were prepared for western blotting. Samples were separated using 10% SDS–PAGE and blotted onto PVDF membranes. The membranes were first incubated with blocking solution (5% skim milk in TBST) and then incubated with the primary antibody in blocking solution. Membranes were washed in TBST three times for 15 min and incubated with a horseradish peroxidase (HRP)-conjugated secondary antibody in blocking solution. The blots were visualized using Chemi-Lumi One, Immobilon, SuperSignal West Femto Maximum Sensitivity Substrate, and LAS-3000 (Fujifilm), followed by analysis using the Multi Gauge Ver3.2 software. β-actin was measured as an internal control. The antibodies used and their dilutions were listed in Key Resources Table.

## In situ hybridization

*Lin28a*<sup>−/−</sup> embryos and Wt littermates were obtained by intercrossing *Lin28a*<sup>+/−</sup> mice. Whole-mount in situ hybridization was performed as described previously (*Yokoyama et al., 2009*); the details of the probe sequence can be obtained from the 'EMBRYS' website (http://embrys.jp/embrys/html/MainMenu.html). Briefly, embryos were fixed in 4% PFA/PBT and dehydrated in a series of increasing MetOH concentrations. Rehydrated samples were bleached with 6% $H_2O_2$ in PBT and treated with 10 µg/mL of Protease K for 10 min at room temperature (RT), stopped with 0.2% glycine, and refixed with 4% PFA/0.2% glutaraldehyde in PBT for 20 min at RT. RNA hybridization was performed at 70°C for more than 14 hr, after prehybridization for 1 hr in hybridization buffer (50% formamide, 5 × SSC, 1% SDS, 50 µg/mL of tRNA, and 50 µg/mL of heparin in RNase-free $H_2O$). Subsequently, embryos were washed three times in wash buffer 1 (50% formamide, 5 × SSC, and 1% SDS in RNase-free $H_2O$) and twice in wash buffer 2 (50% formamide, 2 × SSC, and 5% Chaps in RNase-free $H_2O$). After blocking with 10% sheep serum in TBST for 1 hr at RT, samples were incubated with an anti-DIG-AP Fab fragment antibody and 1% sheep serum in TBST overnight (O/N) at 4°C. After a series of washes with TBST, embryos were equilibrated in alkaline phosphatase buffer (NTMT) and developed with NBT/BCIP solution (Roche). After the color reaction, the embryos were rinsed in TBST several times and postfixed in 4% PFA/PBT at 4°C.

In situ hybridization of sections was performed on Wt and *Lin28a*<sup>−/−</sup> embryos at E12.5, as described previously (*Uchibe et al., 2012*). Embryos were fixed in 4% PFA/PBT, dehydrated in a series of increasing MetOH concentrations, and embedded in paraffin. Sagittal sections (10 µm) were stained with Alcian Blue and Fast Red to outline the pre-vertebrae. Deparaffinized and rehydrated sections were treated with 8 µg/mL of Proteinase K (Roche) in PBS for 10 min, and the reaction was stopped with 0.2% glycine in PBS. After postfixation with 4% PFA, samples were acetylated in acetylation buffer (100 mM triethanolamine, 2.5 mM acetic anhydride; pH was adjusted to 8.0 using HCl). Sections were incubated in prehybridization buffer (50% formamide, 5 × SSC) for 1 hr at 65°C. Subsequently, hybridization was performed O/N at 65°C using an RNA probe for *Hoxc13* in hybridization buffer (50% formamide, 5 × SSC, 10% dextran sulfate, 5 × Denhardt's solution, 0.1 mg/mL of salmon sperm DNA, and 0.25 mg/mL of tRNA). The sections were washed with 0.2 × SSC for 3 hr at 65°C and rinsed with neutralize tagment (NT) buffer (100 mM Tris-HCl, pH 7.5, 150 mM NaCl) for 5 min. After blocking with 10% sheep serum in NT buffer, samples were incubated with an anti-DIG-AP Fab fragment antibody and 1% sheep serum O/N at 4°C. After a series of washes with NT buffer, samples were equilibrated in NTM (100 mM NaCl, 100 mM Tris-HCl, pH 9.5, and 50 mM $MgCl_2$) and developed using an NBT/BCIP solution. After the color reaction, the embryos were counterstained with Fast green.

## Skeletal preparation

Whole-mount skeletal preparations of neonatal mice of each genotype were performed using Alcian Blue and Alizarin Red S staining. For RA treatment, 1 mg/kg of RA was injected intraperitoneally at 7.5 dpc, and the skeletal patterning of each genotype was analyzed at E15.5. The samples were fixed in 100% ethanol (EtOH) for 1–2 days after the majority of the skin and internal organs were removed. The 100% EtOH wash was changed several times. After fixation, the samples were incubated in Alcian Blue solution (0.03% Alcian Blue 8GX, 80% EtOH, and 20% acetic acid) for up to 2 days. The samples were rinsed in distilled water three times and incubated in Alizarin Red Solution (0.01% Alizarin Red S, 1% KOH in $H_2O$) O/N. The samples were treated with discoloring solution (1% KOH, 20% glycerol in $H_2O$) for 4–7 days. The samples were soaked in a series of glycerol/EtOH solutions (20% glycerol, 20% EtOH; 50% glycerol, 50% EtOH) and stored in 100% glycerol.

## Quantitative PCR

Total RNA was isolated from whole embryos (*Figure 3A and B*), or from dissected somites and neural tubes (*Figures 2A*, *3C and G*) at E9.5 using ISOGEN (Nippon Gene), according to the manufacturer's instructions. For SYBR green q-PCR, a complementary DNA (cDNA) was produced using Superscript II reverse transcriptase, 1 µg of total RNA, and an oligo(dT)18 primer. q-PCR analysis was performed using the SYBR Green PCR Master Mix and an ABI 7900HT instrument (Applied Biosystems). *Gapdh* was measured as an internal control to normalize sample differences. The primer sets used for all *Hox* genes were described by *Kondrashov et al., 2011*. The primer sequences used

for other genes were as follows: *Lin28a* Fp1, 5′–CTCGGTGTCCAACCAGCAGT–3′; *Lin28a* Rp1, 5′–CACGTTGAACCACTTACAGATGC–3′; *Lin28a* Fp2, 5′–AGGCGGTGGAGTTCACCTTTAAGA–3′; *Lin28a* Rp2, 5′–AGCTTGCATTCCTTGGCATGATGG–3′; *Cbx2* Fp, 5′–AGGCCGAGGAAACACACAG T–3′; *Cbx2* Rp, 5′–GGAGGAAGAGGACGAACTGC–3′; *Oct3/4* Fp, 5′–GTTTCTGAAG TGCCCGAAGC–3′; *Oct3/4* Rp, 5′–GCGCCGGTTACAGAACCATA–3′; *Nanog* Fp, 5′–ACCTCAGCC TCCAGCAGATG–3′; *Nanog* Rp, 5′–ACCGCTTGCACTTCATCCTT–3′; *Sox2* Fp, 5′–GGCAGC TACAGCATGATGCAGGAGC–3′; *Sox2* Rp, 5′–CTGGTCATGGAGTTGTACTGCAGG–3′; *Gapdh* Fp, 5′–CCTGGTCACCAGGGCTGC–3′; and *Gapdh* Rp, 5′–CGCTCCTGGAAGATGGTGATG–3′.

For microRNAs, cDNAs were produced using the TaqMan MicroRNA Reverse Transcription kit according to the manufacturer's protocol. q-PCR was performed using TaqMan Rodent MicroRNA Array A and B and TaqMan MicroRNA Assays for *let-7a, let-7b, let-7c, let-7d, let-7e, let-7f, let-7g, let-7i, mir-98, mir-10a, mir-10b, mir-196a, mir-196b,* and *RNU6B. RNU6B* was measured as an internal control to normalize sample differences.

## Luciferase assay

The pLuc2 reporter vector was as described previously (*Miyaki et al., 2010*). To create the *let-7* sensor vector, the chemically synthesized *let-7* complementary sequence was annealed and inserted between the EcoRI and XhoI sites. To create the pLuc2-candidate gene 3′UTR vector, the predicted *let-7* target sequence of each genes of 3′UTR was cloned into pLuc2. Fragment containing mutation in *let-7* target sequence were also cloned in pLux2. The miRNA precursor sequence (40 bp) was cloned into pcDNA3.1 and used as an miRNA-expressing vector. Transfection into HEK293T cells was performed using Lipofectamine 2000 or FugeneHD. The transfected cells were incubated for 48 hr, and luciferase activity was determined using the Dual-Glo Luciferase Assay System.

## Chromatin immunoprecipitation

Harvested E9.5 embryos were dissected into somites and neural tubes. Genomic DNA was isolated from the yolk sac and genotyping PCR was performed. Samples were cryopreserved until use. For each assay, ChIP was performed on a pool of 10 embryos. Each antibody (5 µg) was used for immunoprecipitation. The antibodies used for ChIP were listed in Key Resources Table. The frozen samples were cross-linked with 1% formaldehyde in PBS for 10 min at RT. Cross-linking was stopped by adding 100 µL of 1.25 M glycine for 5 min at RT. Samples were washed with PBS and suspended in cell lysis buffer (10 mM Tris-HCl (pH 7.5), 10 mM NaCl, 3 mM MgCl2, 0.5% NP-40, and 1 mM PMSF). Nuclei were collected by centrifugation and resuspended in cell lysis buffer twice. Samples were suspended in 130 µL of nucleus lysis buffer (50 mM Tris-HCl (pH 8.0), 10 mM EDTA (pH 8.0), 1% SDS, and 1 mM PMSF) and transferred into Covaris microTUBEs. The chromatin was sheared by sonication (peak power, 105; duty factor, 5.0; cycles/burst, 200; duration, 10 min). The sheared DNA was diluted in IP dilution buffer (20 mM Tris-HCl (pH 8.0), 2 mM EDTA (pH 8.0), 150 mM NaCl, 1% Triton X-100, 0.1% SDS, and 1 mM PMSF), added to antibody beads, and rotated O/N at 4°C. Precipitated beads with chromatin were washed four times with ChIP wash buffer 1 (20 mM Tris-HCl (pH 8.0), 2 mM EDTA (pH 8.0), 150 mM NaCl, 1% Triton X-100, 0.1% SDS, and 1 mM PMSF) and twice with ChIP wash buffer 2 (20 mM Tris-HCl (pH 8.0), 2 mM EDTA (pH 8.0), 500 mM NaCl, 1% Triton X-100, 0.1% SDS, and 1 mM PMSF). After washing with TE, chromatin was isolated using nucleus lysis buffer at 65°C. The isolated chromatin was de-cross-linked for 6 hr at 65°C. After Proteinase K treatment, DNA was purified using a PCR purification kit (elute in 50 µL of $H_2O$). q-PCR was performed on immunoprecipitated DNA and input DNA and analyzed for the efficiency of immunoprecipitation by each antibody. The primer sequences used for ChIP q-PCR were as follows: ChIP *Hoxa1* Fp, 5′–TGA-GAAAGTTGGCACGGTCA–3′; ChIP *Hoxa1* Rp, 5′–CACTGCCAAGGATGGGGTAT–3′; ChIP *Hoxa2* Fp, 5′–CTCCAAGGAGAAGGCCATGA–3′; ChIP *Hoxa2* Rp, 5′–CGACAGGGGGAAAAGATGTC–3′; ChIP *Hoxa3* Fp, 5′–GTTGTCGCTGGAGGTGGAG–3′; ChIP *Hoxa3* Rp, 5′–GCCAGAGGACGCAG-GAAAT–3′; ChIP *Hoxa4* Fp, 5′–AACGACACCGCGAGAAAAAT–3′; ChIP *Hoxa4* Rp, 5′–GGGAAC TTGGGCTCGATGTA–3′; ChIP *Hoxa5* Fp, 5′–TCCCCCGAATCCTCTGTATC–3′; ChIP *Hoxa5* Rp, 5′–A TTGCATTTCCCTCGCAGTT–3′; ChIP *Hoxa6* Fp, 5′–GTTCGGCCATCCAGAAACA–3′; ChIP *Hoxa6* Rp, 5′–CCCCTCTGCAGGACTGTGAT–3′; ChIP *Hoxa7* Fp, 5′–AGCCTTCACCCGACCTATCA–3′; ChIP *Hoxa7* Rp, 5′–AGCACAGCCTCGTTCTCTCC–3′; ChIP *Hoxa9* Fp, 5′–CCTCCCGGGTTAATTTGTAGC–3′; ChIP *Hoxa9* Rp, 5′–CCCCTGCCTTGGTTATCCTT–3′; ChIP *Hoxa10* Fp, 5′–CCTAGAC

TCCACGCCACCAC–3′; ChIP *Hoxa10* Rp, 5′–GGCTGGAGACAGCTCCTCA–3′; ChIP *Hoxa11* Fp, 5′–AGAGCTCGGCCAACGTCTAC–3′; ChIP *Hoxa11* Rp, 5′–AACTGGTCGAAAGCCTGTGG–3′; ChIP *Hoxa13* Fp, 5′–ACTTCGGCAGCGGCTACTAC–3′; ChIP *Hoxa13* Rp, 5′–CATGTACTTG TCGGCGAAGG–3′; ChIP *Hoxc13* Fp, 5′–CAGGAGACCCAGGCTTAGCA–3′; ChIP *Hoxc13* Rp, 5′–GCATGCGGACACACTTCATT–3′; ChIP *Hoxd12* Fp, 5′–GGAGATGTGTGAGCGCAGTC–3′; ChIP *Hoxd12* Rp, 5′–CTGCCATTGGCTCTCAGGTT–3′.

## Knockdown of *let-7* in ES-like cells

To knockdown *let-7* expression, guide-RNAs targeting the *let-7* family members were constructed. The target sequences of *let-7* family members were as follows: *let-7a-1*, TAGTAGGTTGTATAGTTTT; *let-7a-2* and *let-7c-1*, GGTTGAGGTAGTAGGTTGT; *let-7b*, TAGTAGGTTGTGTGGTTTC; *let-7c-2*, TAGTAGGTTGTATGGTTTT; *let-7d*, TAGTAGGTTGCATAGTTTT; *let-7e*, GTAGGAGGTTGTATAG TTG; *let-7f-1*, TAGTAGATTGTATAGTTGT; *let-7f-2*, TAGTAGATTGTATAGTTTT; *let-7g*, TAGTAG TTTGTACAGTTTG; and *let-7i*, AGGTAGTAGTTTGTGCTGT (see also **Figure 5H**). Four guide-RNA-expressing plasmid vectors and an hCas9 vector (500 ng each) were transfected into $1 \times 10^6$ cells using the Neon transfection system, according to the manufacturer's instructions. Transfected cells were cultured in ES medium containing 0.5 µg/mL of puromycin for 2 days. Each colony was isolated and expanded, followed by PCR and sequence analysis. The primer sequences used for *let-7* genotyping PCR were as follows: *let-7a-1* Fp, 5′–GGCTTATAGCCCAGGTGTATCAT–3′; *let-7a-1* Rp, 5′–ACTTGCCCATTCCCATCATC–3′; *let-7a-2* Fp, 5′–TTCTTATGAACGGCCCGAGT–3′; *let-7a-2* Rp, 5′–CCGTTGATCACCTGTGTTGC–3′; *let-7c-1* Fp, 5′–TGGTAGGCACAGGCCTTTCT–3′; *let-7c-1* Rp, 5′–CAATGTGTGGTTGGCGATCT–3′; *let-7b* Fp, 5′–TTTGCTCGCTGCTAATGGAA–3′; *let-7b* Rp, 5′–GGCCTCATGGACTCATGACA–3′; *let-7c-2* Fp, 5′–GTCTCCCCGTCTCCCCTTAC–3′; *let-7c-2* Rp, 5′–AGGTGCCCTGAAAATGCTGT–3′; *let-7d* Fp, 5′–TTTGGCTTTTGCCAAGATCA–3′; *let-7d* Rp, 5′–TGC TTTCCAAAACTTCCCAGT–3′; *let-7e* Fp, 5′–TGAATTCCTGGGTTCCTTGG–3′; *let-7e* Rp, 5′–TCAAGA TGGCATAGAGACTGCAA–3′; *let-7f-1* Fp, 5′–GATGATGGGAATGGGCAAGT–3′; *let-7f-1* Rp, 5′–CCAAAAGGCCTGGTCCTAGA–3′; *let-7f-2* Fp, 5′–TCTTGTGTGCTTGTCTCCCATT–3′; *let-7f-2* Rp, 5′–CTGAGAACCACTGCCACCAG–3′; *let-7g* Fp, 5′–TGGTGTATTTCTTTTGTTGGGTTG–3′; *let-7g* Rp, 5′–TGAACAACTCCAAGCCTCTCA–3′; *let-7i* Fp, 5′–GGGCCCCGGATGTAAGATGG–3′; and *let-7i* Rp, 5′–CCTCGAGAACGAAACCCAAC–3′. The PCR products were treated with ExoSAP-IT (Affimetrix) and used as templates for direct sequencing. Clones of *let-7* family members with deletions of several nucleotides were selected for further analysis.

Embryoid bodies were produced from each clone, and expression changes of *Hox* genes were analyzed over 3 days. Cells ($1 \times 10^6$) were suspended in 1 mL of DMEM with 10% FBS and plated in low-adhesion culture dishes. After several hours, self-aggregated ES-like cells were resuspended in 10 mL of medium. The medium was changed every other day. RNA isolation and q-PCR analysis are described above.

## Statistical analyses

Two-tailed independent Student's *t*-tests were used to determine all *P* values. Asterisks indicate statistically significant differences (at p<0.05), whereas n.s. indicates an absence of significance.

## Acknowledgements

We thank Dr. Hirohito Shimizu for the technical advice on RA treatment assay, Ms. Moe Tamano for the embryo manipulation, Drs. Satohsi Yamashita and Kazuhiko Nakabayashi for technical advice on ChIp assay, Prof. Mikiko C Siomi for critical and helpful discussion, and Ms. Izumi A Tsune and Dr. Spencer J Spratt for their support in manuscript preparation. We also thank all other Asahara lab members for their support.

# Additional information

## Funding

| Funder | Grant reference number | Author |
|---|---|---|
| Japan Agency for Medical Research and Development | JP17gm0810008 | Hiroshi Asahara |
| National Institute of Arthritis and Musculoskeletal and Skin Diseases | AR050631 | Hiroshi Asahara |
| Japan Society for the Promotion of Science | 15H02560 | Hiroshi Asahara |
| Core Research for Evolutional Science and Technology | JP15gm0410001 | Hiroshi Asahara |
| Japan Society for the Promotion of Science | 13J00119 | Tempei Sato |
| Japan Society for the Promotion of Science | 26113008 | Hiroshi Asahara |
| National Institute of Arthritis and Musculoskeletal and Skin Diseases | AR065379 | Hiroshi Asahara |
| Japan Society for the Promotion of Science | 26113008 | Hiroshi Asahara |
| Japan Society for the Promotion of Science | 15K15544 | Hiroshi Asahara |

The funders had no role in study design, data collection and interpretation, or the decision to submit the work for publication.

## Author contributions

Tempei Sato, Hiroshi Asahara, Conceptualization, Resources, Data curation, Supervision, Funding acquisition, Validation, Investigation, Methodology, Project administration; Kensuke Kataoka, Conceptualization, Data curation, Funding acquisition, Validation, Investigation, Methodology; Yoshiaki Ito, Shigetoshi Yokoyama, Hiroe Ueno-Kudoh, Investigation; Masafumi Inui, Conceptualization, Investigation; Masaki Mori, Satoru Takahashi, Conceptualization, Supervision, Investigation; Keiichi Akita, Shuji Takada, Supervision, Investigation

## Author ORCIDs

Tempei Sato https://orcid.org/0000-0002-8966-3374
Shigetoshi Yokoyama http://orcid.org/0000-0003-4175-0548
Masafumi Inui http://orcid.org/0000-0003-4720-007X
Keiichi Akita http://orcid.org/0000-0002-2927-2937
Hiroshi Asahara https://orcid.org/0000-0002-5215-8745

## Ethics

Animal experimentation: All animal experiments were performed in accordance with protocols approved by the Institutional Animal Care and Use Committee of the National Research Institute for Child Health and Development (permit numbers: 2004-003, 2014-001).

## Decision letter and Author response

Decision letter https://doi.org/10.7554/eLife.53608.sa1
Author response https://doi.org/10.7554/eLife.53608.sa2

## Additional files

### Supplementary files
- Supplementary file 1. Survival rate of *Lin28a* mutant mice at various stages.
- Transparent reporting form

### Data availability
All data generated or analysed during this study are included in the manuscript and supporting files.

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
