## [Decision Letter]

**Acceptance summary:**

The paper demonstrates a role for the temporal regulatory microRNA let-7, and its mutual target, Lin-28, in skeletal anterior-posterior patterning. These findings reinforce the importance of precise timing and dosage of gene activity in achieving the proper spatial patterning of Hox gene activity.

**Decision letter after peer review:**

Thank you for submitting your article "Lin28a/let-7 Pathway Modulates the Hox Code via Polycomb Regulation during Axial Patterning in Vertebrates" for consideration by *eLife*. Your article has been reviewed by three peer reviewers, and the evaluation has been overseen by a Reviewing Editor and Clifford Rosen as the Senior Editor.

The reviewers have discussed the reviews with one another and the Reviewing Editor has drafted this decision to help you prepare a revised submission.

Summary:

Sato et al. demonstrate a role in skeletal anterior-posterior patterning for the temporal regulatory microRNA let-7, and its mutual target, Lin-28. This is an important advance, because it reinforces the importance of precise timing and dosage of gene activity in achieving the proper spatial patterning of Hox gene activity. In addition the authors show that the action of Lin-28a/let-7 in axial patterning very likely reflects the regulation of the PcG gene Cbx2, thus linking these temporal regulators to the process of setting up the chromatin state of Hox genes. They exploited an ES cell differentiation model to critically test for whether let-7 family microRNAs mediate the effects on Hox gene expression of Lin-28a loss-of-function, and also to test for whether the let-7 complementary sites in the Cbx2 3' UTR are required to mediate the action of let-7 in this context.

Essential revisions:

This work is a technically thorough investigation of the phenotype of Lin28 mutant mice. However, the reviewers point out that the relatively single-sighted focus on Hox genes likely represents a 'miss' in interpreting the phenotype. There are many inconsistencies in this focus. Among them is using the term 'homeotic transformations' is technically acceptable perhaps, but the authors are really reporting a shortening of the axis with perhaps a 'perturbation' of C1/C2. These are mild 'transformations' to begin with and not really homeotic. The more substantial issue with this version of the manuscript is that it does not take into account a publication from February from Moises Mallo's laboratory (Aires, et al., Dev Cell), which closely examined a network of factors, including Lin28 in during tailbud transition. It seems the authors are not aware of this work that profoundly impacts the interpretation of this phenotype. It is critical that these authors carefully read this paper and rewrite theirs and resubmit with what seems likely to be a re-interpretation of their interesting phenotype – or, if not that, their interpretation in light of this work.

Reviewer #1:

1) It is not clear what the RA experiment adds to the story. As it stands, the argument is based on saying, "It is known that Hox genes are modulated by retinoic acid…" and then, "Therefore we administered…". The logic of the experiment needs to be more explicitly laid out. Also, the conclusions are not clearly linked to the result. At the end of the paragraph, the conclusion is drawn that the sensitivity of Lin-28a+/- to RA indicates that, “Lin-28a acts upstream of the Hox genes." But it is not immediately apparent that the observed synergy between Lin-28a+/- and RA might not be consistent with alternative interpretations, such as parallel pathways.

2) "To support this concept further…." is not really a satisfactory rationale for this genetic interaction experiment. The logic should be spelled out more explicitly. There is a related problem with the statement of the conclusions of the experiment, in the final sentence of the paragraph. "…these results suggest that the Lin-28a/let-7 reciprocal feedback regulates Cbx2 expression…." The authors need to tell us exactly why the synergy between Lin-28+/- and Cbx2+/- means that Lin-28a regulates Cbx2. In particular, does this result necessarily rule out the possibility that Cbx2 and Lin-28a could regulated skeletal patterning by parallel pathways? Also, the statement that "…this pathway is required for skeletal patterning…" is a bit too strong, as "required for" would imply strong loss-of-function phenotypes. More appropriate would be something like, "…this pathway contributes to the regulation of proper skeletal patterning…"

3) Subsection “Let-7 knockdown rescues Hox gene dysregulation in Lin28a^–/–^ cells”, third paragraph: The data in Figure 5I needs to be quantitated; i.e., band intensities scanned. This is also the case for Figure 3—figure supplement 2. In order to argue that there was indeed a change in protein level, replicate samples need to be measured quantitatively, and statistical tests should be applied.

4) Subsection “Let-7 knockdown rescues Hox gene dysregulation in Lin28a^–/–^ cells”, last paragraph: Referring to Figure 5L this point is confusing, because the depiction of the model in Figure 5L does not really account for how reduced Lin-28a would result in posterior transformations in cell fate, which is what the results indicate. Now, in the Discussion section (see below), the model is much better justified by taking into account what is known about the importance of timing in the establishment of proper Hox gene expression patterns. By the way, one source of confusion (at least for this reader) is the fact that the down regulation of Lin-28a during development occurs in a posterior-to-anterior direction; this led me to think in terms of the differential temporal regulation of posterior vs. anterior Hox genes by Lin-28a during e9.5-e10-5. But, unless I am still confused, it seems that the axial sweep of Lin-28a down regulation is not in itself terribly germane to the model.

5) Discussion, first paragraph: Here the model is made much more understandable by linking hox gene expression patterns to the timing of their repression or activation. Another key point is made where it is stated that "PcG genes should be repressed prior to the initiation of Hox gene expression to open the chromatin….". The diagrams shown in Figure 5L does not effectively convey these important temporal elements of the authors' model.

6) Discussion, second paragraph: See comments above about the need to explicate the logic for how the genetic interaction results support the conclusions. Don't get me wrong; I agree that these results support the model, but the logic is just not expressed in the narrative. Note, however that there are always caveats about interpreting genetic interactions too definitively. For example, the authors' results are also consistent with Lin-28a and Cbx2 acting in the same linear pathway, or in parallel pathways.

7) Figure 5 legend: As discussed above, I do not find the diagram in Panel L terribly helpful in understanding precisely what is the authors' model for how the developmental dynamics of Lin-28a/let-7 activity impacts hox gene spatial expression patterns. It is clear from the diagram that the "Time dependent down-regulation of Lin-28a" is important, and emphasis is placed on the fact that this down-regulation occurs progressively from the anterior towards the posterior of the embryo. However, there is no depiction of how those dynamics are connected to Hox patterning. In particular, the model depicts a transition from broad PcG expression at 9.5 to a posterior-enriched PcG expression pattern at 10.5, which seems to imply that the repression of Lin-28a by let-7 is similarly from anterior to posterior; but I'm not certain if that is the authors' intent. Also, the model also seems to suggest that the Lin-28a/let-7 circuit could be entirely responsible for the posterior-anterior down-regulation of PcG gene expression, but of course that is not the case.

Reviewer #2:

Introduction, last paragraph – as stated above, the phenotype is somewhat mild (except for the tail, i.e. Aires, et al) and is more accurately a shortening of the axis, not a homeotic transformation. Likewise, one would have to hypothesize a 'dysregulation of Hox genes, but the authors see an increase later. As shown by clear genetic work (Wellik and Capecchi, 2003, McIntyre, et al., 2007), Hox genes are not very sensitive to dosage! There is no reason to hypothesize that increasing expression – except for Hox13 (Mallo work) – would lead to changes observed here. The interpretation is highly consistent with Aires, et al., in axis elongation and conversion to tail bud!

Subsection “Hox genes are dysregulated in Lin28a^–/–^ mice”: 'global transformation' is a very liberal interpretation of phenotype. It is mainly tail!

The RA experiments add little in this reviewer's view.

The let7 activity becomes more interesting in light of Aires, et al. and its effects on lin28 are highly interesting (if also supporting the strong conservation one might expect from previous work). The ES cell work is not highly compelling in light of Aires, et al., but could perhaps be used in light of this context…

The data supporting Cbx2 mutants as having 'homeotic transformations' is less supported than for Lin28. This statement should be highly modified or removed. It is very unclear why authors did not look at double mutants. One suspects this may not support 'Hox' part of story, but in light of Aires, et al., this phenotype may be highly interesting and allow support of an alternative interpretation. This reviewer is certainly curious about what happens! And a reinterpretation in general may be warranted here especially.

Subsection “Lin28a/let-7 pathway modulates PRC1 occupancy at posterior Hox loci” – very odd references to Hox mutants – the paralog mutants that establish clear homeosis as predicted from *Drosophila* work should be discussed here (Horan, van den Akker, Wellik, McIntyre).

Discussion and Reinterpretation of let7 experiments should be re-worked in light of Aires.

Reviewer #3:

1) Lin28a is ubiquitously expressed in embryonic development. Hox genes as well as Cbx2 have been shown to regulate AP patterning and other developmental processes such as limb patterning. Are there any other developmental defects observed in the Lin28a-/- mice that also support the role of Lin28a in regulating Cbx2 and Hox codes?

2) Most importantly, Figure 4H indicates that only 5' Hox genes and posterior regions should be affected in the Lin28a-/- mice, but homeotic transformations were seen in the cervical, thoracic and lumbosacral regions where Hoxc13 and Hoxd12 were not expressed.

3) The expression of Hox genes should be examined in more detail in both WT and mutant embryos at the same stages. In Figure 2B, how were the expression domains of Hoxc13 and Hoxd12 demarcated?

4) In Figure 2—figure supplement 1, comparison of Hox gene expression patterns should be done at the same stages with similar somite numbers and the expression domain should be properly demarcated with somite "landmarks".

5) The readouts of genetic interaction experiments by crossing to Cbx2 mutants and RA treatment were not robust. If posterior regions were more affected, homeotic transformation in the lumbosacral regions should be shown.

6) There seems to be some discrepancy in the regulation of Hox genes by Lin28a in vitro and in vivo. For instance, Hoxa11 expression was altered in vitro, but not in the somites in development by WISH (Figure 4E).

---

## [Author Response]

Essential revisions:This work is a technically thorough investigation of the phenotype of Lin28 mutant mice. However, the reviewers point out that the relatively single-sighted focus on Hox genes likely represents a 'miss' in interpreting the phenotype. There are many inconsistencies in this focus. Among them is using the term 'homeotic transformations' is technically acceptable perhaps, but the authors are really reporting a shortening of the axis with perhaps a 'perturbation' of C1/C2. These are mild 'transformations' to begin with and not really homeotic.

We agree with the reviewer’s comment that our observations of Lin28a-/- mice are rather mild 'transformations' not apparent 'homeotic transformations'. To avoid being potentially misleading in the interpretation of the phenotype, we have carefully revised the specific terms for each observed phenotype. As for skeletal patterning defects of C1/C2, we rephrase 'homeotic transformations' using other words such as 'skeletal transformation'; Abstract, Introduction, Results, Discussion and Figure 1 legend.

The more substantial issue with this version of the manuscript is that it does not take into account a publication from February from Moises Mallo's laboratory (Aires, et al., Dev Cell), which closely examined a network of factors, including Lin28 in during tailbud transition. It seems the authors are not aware of this work that profoundly impacts the interpretation of this phenotype. It is critical that these authors carefully read this paper and rewrite theirs and resubmit with what seems likely to be a re-interpretation of their interesting phenotype – or, if not that, their interpretation in light of this work.

We appreciate the reviewer’s advice. Although we had noted, and cited,Dr. Mallo's report (Aires, et al., Dev Cell) in the Discussion of the original manuscript, we should have discussed the comparison between their finding and our observation in more depth. This has now been included in the revised manuscript, in the Abstract, the Introduction, the Results and extensively in the Discussion.

Reviewer #1:1) It is not clear what the RA experiment adds to the story. As it stands, the argument is based on saying, "It is known that Hox genes are modulated by retinoic acid…" and then, "Therefore we administered…". The logic of the experiment needs to be more explicitly laid out. Also, the conclusions are not clearly linked to the result. At the end of the paragraph, the conclusion is drawn that the sensitivity of Lin-28a+/- to RA indicates that, “Lin-28a acts upstream of the Hox genes." But it is not immediately apparent that the observed synergy between Lin-28a+/- and RA might not be consistent with alternative interpretations, such as parallel pathways.

We thank the reviewer for this comment and have better explained the logic of the experiment in the revised manuscript (subsection “Hox genes are dysregulated in Lin28a–/– mice). Specifically, we hypothesized that the patterning defects of vertebrae observed in *Lin28a*^–/–^ mice were caused by the perturbation of *Hox* gene expression. To test that, we investigated the effects of perturbation of Hox gene expression by RA on skeletal pattern formation in Lin28a mutants. We conclude more cautiously by now saying that the results show that Hox gene perturbation by RA administration enhanced the Lin28a+/- and -/- phenotypes. In particular, since RA treated Lin28a+/- shows the same phenotype as untreated Lin28a-/-, we consider that there is a possibility that dysregulation of *Hox* genes might be responsible for the skeletal patterning defects in *Lin28a*^–/–^mice (Figure 2E).

2) "To support this concept further…." is not really a satisfactory rationale for this genetic interaction experiment. The logic should be spelled out more explicitly. There is a related problem with the statement of the conclusions of the experiment, in the final sentence of the paragraph. "…these results suggest that the Lin-28a/let-7 reciprocal feedback regulates Cbx2 expression…." The authors need to tell us exactly why the synergy between Lin-28+/- and Cbx2+/- means that Lin-28a regulates Cbx2. In particular, does this result necessarily rule out the possibility that Cbx2 and Lin-28a could regulated skeletal patterning by parallel pathways? Also, the statement that "…this pathway is required for skeletal patterning…" is a bit too strong, as "required for" would imply strong loss-of-function phenotypes. More appropriate would be something like, "…this pathway contributes to the regulation of proper skeletal patterning…"

We thank the reviewer for this helpful comment and apologise for the lack of clarity in the original version, and appearing to overinterpret the data. We now better explain the rationale, specifically that Cbx2 is a polycomb protein that epigenetically regulates Hox genes, and cite a new reference showing that ablation of Cbx2 shows skeletal patterning defects (Core et al., 1997), leading us to speculate that decreased expression of Cbx2 might cause the abnormal skeletal formation in *Lin28a*^-/-^mice. We also more carefully describe the *Cbx2* homozygous mutant phenotype, and describe our reasoning why dysregulation of *Cbx2* might be responsible for the phenotype of *Lin28a*^–/–^mice, as well as being more cautious in our interpretations, for example by concluding that this pathway contributes to the regulation of proper skeletal patterning, as suggested. See subsection “Lin28a regulates Cbx2 expression via let-7 repression”.

3) Subsection “Let-7 knockdown rescues Hox gene dysregulation in Lin28a^–/–^ cells”, third paragraph: The data in Figure 5I needs to be quantitated; i.e., band intensities scanned. This is also the case for Figure 3—figure supplement 2. In order to argue that there was indeed a change in protein level, replicate samples need to be measured quantitatively, and statistical tests should be applied.

We have now quantified these data using imageJ, and added them to the revised Figure 5I and Figure 3—figure supplement 2.

4) Subsection “Let-7 knockdown rescues Hox gene dysregulation in Lin28a^–/–^ cells”, last paragraph: Referring to Figure 5L this point is confusing, because the depiction of the model in Figure 5L does not really account for how reduced Lin-28a would result in posterior transformations in cell fate, which is what the results indicate. Now, in the Discussion section (see below), the model is much better justified by taking into account what is known about the importance of timing in the establishment of proper Hox gene expression patterns. By the way, one source of confusion (at least for this reader) is the fact that the down regulation of Lin-28a during development occurs in a posterior-to-anterior direction; this led me to think in terms of the differential temporal regulation of posterior vs. anterior Hox genes by Lin-28a during e9.5-e10-5. But, unless I am still confused, it seems that the axial sweep of Lin-28a down regulation is not in itself terribly germane to the model.

We apologize for the difficulty of understanding the contents of the figure. We revised the model with your helpful suggestions (Figure 5L).

5) Discussion, first paragraph: Here the model is made much more understandable by linking hox gene expression patterns to the timing of their repression or activation. Another key point is made where it is stated that "PcG genes should be repressed prior to the initiation of Hox gene expression to open the chromatin….". The diagrams shown in Figure 5L does not effectively convey these important temporal elements of the authors' model.

We thank the reviewer for this helpful comment. We have revised the diagram with the temporal regulation of gene expression as described above.

6) Discussion, second paragraph: See comments above about the need to explicate the logic for how the genetic interaction results support the conclusions. Don't get me wrong; I agree that these results support the model, but the logic is just not expressed in the narrative. Note, however that there are always caveats about interpreting genetic interactions too definitively. For example, the authors' results are also consistent with Lin-28a and Cbx2 acting in the same linear pathway, or in parallel pathways.

We thank the reviewer for this helpful comment and have revised the manuscript to address this concern, Discussion, third paragraph.

7) Figure 5 legend: As discussed above, I do not find the diagram in Panel L terribly helpful in understanding precisely what is the authors' model for how the developmental dynamics of Lin-28a/let-7 activity impacts hox gene spatial expression patterns. It is clear from the diagram that the "Time dependent down-regulation of Lin-28a" is important, and emphasis is placed on the fact that this down-regulation occurs progressively from the anterior towards the posterior of the embryo. However, there is no depiction of how those dynamics are connected to Hox patterning. In particular, the model depicts a transition from broad PcG expression at 9.5 to a posterior-enriched PcG expression pattern at 10.5, which seems to imply that the repression of Lin-28a by let-7 is similarly from anterior to posterior; but I'm not certain if that is the authors' intent. Also, the model also seems to suggest that the Lin-28a/let-7 circuit could be entirely responsible for the posterior-anterior down-regulation of PcG gene expression, but of course that is not the case.

We thank the reviewer for helpful comments and suggestions. We have revised the diagram to accommodate the temporal regulation of gene expression as described above (response for reviewer’s comment #4), and to explain our hypothesis that upregulation of let-7 in Lin28a-/- mice leads to decreased Cbx2 expression and subsequent PRC1 occupancy at Hox loci, which causes the disruption of the “Hox code.” *Lin28a* expression gradually diminishes from anterior to posterior during embryogenesis, however, it is not implies that Cbx2 expression is also downregulated from anterior to posterior.

Reviewer #2:Introduction, last paragraph – as stated above, the phenotype is somewhat mild (except for the tail, i.e. Aires, et al) and is more accurately a shortening of the axis, not a homeotic transformation. Likewise, one would have to hypothesize a 'dysregulation of Hox genes, but the authors see an increase later. As shown by clear genetic work (Wellik and Capecchi, 2003, McIntyre, et al., 2007), Hox genes are not very sensitive to dosage! There is no reason to hypothesize that increasing expression – except for Hox13 (Mallo work) – would lead to changes observed here. The interpretation is highly consistent with Aires, et al., in axis elongation and conversion to tail bud!

We appreciate the comment. To avoid a potentially misleading description of the phenotype, we have now carefully replaced every instance of ‘homeotic transformation’, and used other wording, including 'skeletal transformation' and ‘skeletal patterning’: Abstract, Introduction, Results, Discussion and Figure 1 legend.

Although we indeed noted and citedDr. Mallo's report (Aires, et al., Dev Cell) in the Discussion of our original manuscript, we appreciate now that we should have discussed the comparison between their findings and our observations in more depth. This has now been included in the revised manuscript, in the Abstract, the Introduction, the Results and extensively in the Discussion.

Subsection “Hox genes are dysregulated in Lin28a^–/–^ mice”: 'global transformation' is a very liberal interpretation of phenotype. It is mainly tail!The RA experiments add little in this reviewer's view.

We have revised the manuscript to describe that *Lin28a*^–/–^mice exhibited multiple transformations, and have tried to better rationalize and explain the RA experiment. Briefly, we used the RA experiment to support the model that *Lin28a* acts upstream of the *Hox* genes, and that dysregulation of *Hox* genes might be responsible for the phenotype of *Lin28a*^–/–^mice (Figure 2E). We therefore think including it in the manuscript is justified and does add value.

The let7 activity becomes more interesting in light of Aires, et al. and its effects on lin28 are highly interesting (if also supporting the strong conservation one might expect from previous work). The ES cell work is not highly compelling in light of Aires, et al., but could perhaps be used in light of this context…

As mentioned above, we have revised the Discussion to include more in-depth discussion of the previous findings by Aires et al. Briefly, they showed that Lin28 and Hox13 had opposite functions in tail bud proliferation, suggesting that the balance of the expression of those two genes, which might be regulated by GDF signaling, is one of the determinants of tail length. Our results revealed the epigenetic inhibition of HoxPG13 by the Lin28a/let-7/Cbx2 pathway, which might be one of the mechanisms that explains the antagonistic function of Lin28a and HoxPG13 in axial elongation as well as in skeletal patterning. In Figure 5, we used embryoid bodies as a model and tested whether that Hox gene dysregulation could be rescued by knockdown of let-7 or Cbx2 3’UTR mutation. Unfortunately, we tried to test Hox13 expression as well, however, unfortunately, Hox13 was below the detection limit and could not be verified in this differentiation assay. However, we could rescue the gene expression Hox11 and Hox12 in Lin28a-/- cells, and these results suggest a new role of the Lin28a/let-7 pathway in the modulation of the “Hox code.”

The data supporting Cbx2 mutants as having 'homeotic transformations' is less supported than for Lin28. This statement should be highly modified or removed. It is very unclear why authors did not look at double mutants. One suspects this may not support 'Hox' part of story, but in light of Aires, et al., this phenotype may be highly interesting and allow support of an alternative interpretation. This reviewer is certainly curious about what happens! And a reinterpretation in general may be warranted here especially.

Both Lin28a-/- mice (in this study) and Cbx2-/- mice (Coré et al., 1997; Katoh-Fukui et al., 1998, in this study) exhibited skeletal transformations in the C1/C2 region, T13/L1 region, and L6/S1 region. In the case of Lin28a;Cbx2 double mutant mice, ablation or truncation of the 13^th^ pair of ribs were observed (Figures 3I and 3J), although the *Lin28*+/– and *Cbx2*+/– single mutants did not show any obvious phenotypic irregularities. In the lumbosacral region, no additional phenotypes were observed in the double mutants, since almost all of the Lin28a+/- mice showed the same phenotype as the Lin28a-/- mice, which had only five lumbar vertebrae (Table 1). In caudal vertebrae, there were also no additional phenotypes in Lin28a;Cbx2 double mutant mice. Unfortunately, we could not obtain and analyze *Lin28a*/*Cbx2* double KO mice. This is probably because most *Lin28a*^-/-^ exhibited embryonic lethality, and *Cbx2*^-/-^also showed perinatal or postnatal lethality.

Recently, the mechanism of tail bud elongation regulation by Lin28a and Hox13 has been reported (Aires et al., 2019). According to Aires et al., Lin28a/b genes promote and Hox13 genes restrict tail bud progenitor expansion downstream of Gdf11. Elevated Hox13 expression also directly or indirectly suppresses Lin28, leading to shortening of the tail. Since the expression of Hox13 is increased in Lin28a-/-, shortening of the tail is considered to be due to abnormal expression of Hox13. Cbx2 epigenetically suppresses Hox13, and in fact, histone modification of the Hox13 loci was altered in Lin28a-/- mice.　Thus, Lin28a/let-7-mediated Hox13 regulation by Cbx2 might be required for tail bud elongation and vertebrae formation. However, Cbx2-deficient mice did not show caudal vertebral defects and there was no effect on the tail of the Lin28a/Cbx2 double mutant. These results indicate that Hox13 regulation by Lin28a, via both Cbx2-dependent and -independent pathways, might be involved in tail bud progenitor expansion.

Subsection “Lin28a/let-7 pathway modulates PRC1 occupancy at posterior Hox loci” – very odd references to Hox mutants – the paralog mutants that establish clear homeosis as predicted from *Drosophila* work should be discussed here (Horan, van den Akker, Wellik, McIntyre).

We apologize for the omission. We now cite these papers in the Introduction.

Discussion and Reinterpretation of let7 experiments should be re-worked in light of Aires.

We have reworked the manuscript as described above.

Reviewer #3:1) Lin28a is ubiquitously expressed in embryonic development. Hox genes as well as Cbx2 have been shown to regulate AP patterning and other developmental processes such as limb patterning. Are there any other developmental defects observed in the Lin28a-/- mice that also support the role of Lin28a in regulating Cbx2 and Hox codes?

As the reviewer points out, Hox mutant mice show multiple phenotypes, including skeletal patterning defects in both vertebrae and appendages. We found abnormal expression of posterior Hox genes such as Hox12 and Hox13, which are critical for limb patterning, however, there were no abnormalities on limb development. These results indicate the existence of different mechanisms for the regulation of posterior Hox genes by Lin28a in the vertebrae and limb bud.

Previously two independent Cbx2 mutants were generated by different groups (Coré et al., 1997; Katoh-Fukui et al., 1998). Both groups reported that Cbx2 KO mice exhibited skeletal patterning defects, neonatal and postnatal lethality with abnormal body weight. Moreover, Cbx2 KO mice showed male to female sex reversal (Katoh-Fukui et al., 1998), abnormal lymphocyte differentiation, and decreased proliferation of splenocytes and fibroblasts (Coré et al., 1997). Lin28a-/- mice showed skeletal defects similar to that observed in Cbx2 KO mice, however, the other phenotypes, such as sex determination defects, were not found in Lin28a-/- mice. We have not examined lymphocytes and fibroblasts, but in Lin28a-/- ES-like cells that we established in this report, there were no abnormalities in cell proliferation.

2) Most importantly, Figure 4H indicates that only 5' Hox genes and posterior regions should be affected in the Lin28a-/- mice, but homeotic transformations were seen in the cervical, thoracic and lumbosacral regions where Hoxc13 and Hoxd12 were not expressed.

We thank the reviewer for this important comment. Although we do not have a clear answer yet, we considered some possible explanations for the cause of the phenotype in the cervical, thoracic and lumbosacral regions: it is possible that the skeletal pattern was perturbed due to an abnormality in the cell fate determination of the mesenchymal system and the nervous system.　We have revised the manuscript to include a discussion of this issue in the Discussion (second paragraph).

In addition, we have revised and simplified Figure 4H to enhance the reader’s understanding.

3) The expression of Hox genes should be examined in more detail in both WT and mutant embryos at the same stages. In Figure 2B, how were the expression domains of Hoxc13 and Hoxd12 demarcated?

WISH was performed using wild-type and mutants of the same developmental stage as possible. The distance from the hind limb to the anterior expression domain on the body axis was compared.

4) In Figure 2—figure supplement 1, comparison of Hox gene expression patterns should be done at the same stages with similar somite numbers and the expression domain should be properly demarcated with somite "landmarks".

We have tried co-staining with somite markers and Hox genes, but there were technical difficulties using mouse embryos. Although the expression region was examined while confirming the somites on the photograph, no anterior shift of the expression domain was confirmed in genes except for Hoxc13 and Hoxd12.

5) The readouts of genetic interaction experiments by crossing to Cbx2 mutants and RA treatment were not robust. If posterior regions were more affected, homeotic transformation in the lumbosacral regions should be shown.

At first, there were no abnormalities in the lumbosacral and caudal vertebrae in Lin28a;Cbx2 double mutant mice and RA treated Lin28a+/- embryos.

In the RA treatment experiment, no abnormalities were found in the lumbosacral and caudal vertebrae in Lin28a+/- embryos with RA. This might be due to the concentration of RA administered, but it could not be verified at higher concentrations because the fetuses died regardless of genotype.

Both Lin28a-/- mice (in this study) and Cbx2-/- mice (Coré et al., 1997; Katoh-Fukui et al., 1998, in this study) exhibited skeletal transformations in the C1/C2 region, T13/L1 region, and L6/S1 region. In the case of Lin28a;Cbx2 double mutant mice, ablation or truncation of the 13^th^ pair of ribs was observed, although the *Lin28*+/– and *Cbx2*+/– single mutants did not show any obvious phenotypic irregularities (Figures 3I and 3J). In the lumbosacral region, no additional phenotypes were observed in the double mutants, since almost all of the Lin28a+/- mice showed the same phenotype as the Lin28a-/- mice, which had only five lumbar vertebrae (Table 1). In caudal vertebrae, there were also no additional phenotypes in Lin28a;Cbx2 double mutant mice.

Recently, the mechanism of tail bud elongation regulation by Lin28a and Hox13 has been reported (Aires et al., 2019). According to Aires et al., Lin28a/b genes promote and Hox13 genes restrict tail bud progenitor expansion in downstream of Gdf11. Elevated Hox13 expression also directly or indirectly suppresses Lin28, leading to shortening of the tail. Since the expression of Hox13 is increased in Lin28a-/-, shortening of the tail was considered to be due to abnormal expression of Hox13. Cbx2 epigenetically suppresses Hox13, and in fact, histone modification of the Hox13 loci was altered in Lin28a-/- mice.　Thus, Lin28a/let-7-mediated Hox13 regulation by Cbx2 might be required for tail bud elongation and vertebrae formation. However, Cbx2-deficient mice did not show caudal vertebral defects and there was no effect in the tail of Lin28a/Cbx2 double mutant. These results indicate that Hox13 regulation by Lin28a occurs via both a Cbx2-dependent and -independent pathway, and might be involved in tail bud progenitor expansion. The above points are added to the revised Discussion section.

6) There seems to be some discrepancy in the regulation of Hox genes by Lin28a in vitro and in vivo. For instance, Hoxa11 expression was altered in vitro, but not in the somites in development by WISH (Figure 4E).

No change in the expression domain was observed except for Hox12 and Hox13 by WISH, however, the up-regulation of many Hox genes, such as Hoxa11, was found in embryonic somites and neural tubes by qPCR. In fact, ChIP analyses showed a decrease in the repressive histone modification, H2AK119ub, confirming the increased expression of the Hox gene in embryonic somites and neural tubes. So, in terms of gene expression levels, we believe there is no contradiction between the in vivo and in vitro experiments.